



# GGCP10: A Global Gridded Crop Production Dataset at 10km Resolution from 2010 to 2020

Xingli Qin[1], Bingfang Wu[1 2 *], Hongwei Zeng[1], Miao Zhang[1], Fuyou Tian[1]

[1]State Key Laboratory of Remote Sensing Science, Aerospace Information Research Institute, Chinese Academy of Sciences, Beijing 100101, China
[2]College of Resources and Environment, University of Chinese Academy of Sciences, Beijing 100049, China

*Correspondence to*: (wubf@aircas.ac.cn)

Spatial-temporal distribution information on global crop production is of is crucial for studying global food security and promoting sustainable agricultural development. However, the presently available datasets related to this subject are characterized by coarse resolution and discontinuous time spans. To tackle these problems, we have integrated multiple data sources, including statistical data, gridded production data, agroclimatic indicator data, agronomic indicator data, global land surface satellite products and ground data, to develop a data-driven crop production spatial allocation model, and generated the first global temporally continuous 10km resolution gridded production dataset of four major crops (maize, wheat, rice and soybean) from 2010 to 2020 (Global gridded crop production dataset at 10km, GGCP10). A set of data-driven models were trained based on agro-ecological zones to achieve accurate predictions of crop production for different agricultural regions. The performance of the models is demonstrated by the cross-validation results. The accuracy and reliability of GGCP10 have been evaluated from various perspectives using gridded, survey and statistical data. GGCP10 can reveal the spatial-temporal distribution patterns of global crop production and contribute to the understanding of the mechanisms driving changes in crop production. GGCP10 provides crucial data support for research on global food security and sustainable agricultural development. The GGCP10 dataset is available on Harvard Dataverse: https://doi.org/10.7910/DVN/G1HBNK(Qin et al., 2023).



# 1 Introduction

Crop production information plays a critical role in global food security and sustainable agricultural development (Wu et al., 2022; Ray et al., 2012). The four major crops, namely maize, wheat, rice, and soybean, contribute over 64% of the world's caloric intake (Ray et al., 2012). The increased demand for food, coupled with global climate change and population growth, puts immense pressure on countries to secure their food supplies(Gil et al., 2019; Hinz et al., 2020). Thus, there is a growing need to gain insight into food production distribution for sustainable agriculture (Foley et al., 2011; Clark et al., 2020; Myers et al., 2017). Therefore, it is critical to develop a long-term, high-precision dataset of global crop production distribution for research on food production and consumption, policy-making, optimizing resource use, and planning for sustainable agricultural development (Dempewolf et al., 2014).

Currently available global crop production datasets include SPAM (Yu et al., 2020) which covers the years 2000, 2005 and 2010, M3-Crops (Monfreda et al., 2008) which covers the year 2000, GDHY (Iizumi and Sakai, 2020) which covers the period from 1995 to 2005 at a five-year interval, GGCMI (Müller et al., 2019) which covers the period from 1901 to 2012 at a ten-year interval, and GAEZ (Grogan et al., 2022) which covers the years 2010 and 2015. Nonetheless, the different research purposes and technical limitations of these datasets result in insufficient temporal and spatial resolution and coverage. Furthermore, the lack of temporal continuity and timeliness of the data fails to capture the effects of drastic global climate changes that occurred in the past decade (Zhang et al., 2016; Kukal and Irmak, 2018). Therefore, there remains a global shortage of long-term, high-resolution, and gridded crop production datasets.

The era of remote sensing big data has produced a wealth of global observation data, which offers new opportunities to address the spatial distribution of crop production. These massive and diverse remote sensing data contain rich information related to crop production, such as climate, land cover and vegetation growth conditions(Benami et al., 2021; Ahmad et al., 2021). Additionally, ground information such as soil characteristics and topographic conditions, also serve as essential references for estimating crop production. Machine learning techniques have exhibited solid performance in predicting



crop yields and production(Cai et al., 2019; Ji et al., 2022; Li et al., 2023), thereby revealing the deep correlations between crop production and various observation indicators in recent years. Hence, integrating information from multiple sources and using machine learning models to uncover the intrinsic relationships between crop production and observation indicators to obtain accurate spatial distributions of crop production is a feasible approach(Zhang et al., 2019; Han et al., 2020).

A global gridded dataset of maize, wheat, rice, and soybean production was constructed at a 10 km resolution from 2010 to 2020 in this study(Qin et al., 2023). Developing this dataset involved utilizing a data-driven spatial production allocation model that incorporated multiple source datasets, and it was rigorously examined through pre-processing and consistency checks to ensure the accuracy and reliability of the data. The dataset can significantly support monitoring global food security and promoting sustainable development by providing reliable data.

## 2 Data and Methods

### 2.1 Data

To construct the production allocation model, we collected data from multiple sources, including FAO statistical data, GAEZ+ 2015 annual crop data, CropWatch crop phenology data, CropWatch global eco-agricultural zoning, Harmonized World Soil Database (HWSD) soil texture data, CropWatch irrigated land distribution data, latitude and longitude, topographic data, CropWatch agroclimatic indicators, CropWatch agronomic indicators and Global Land Surface Satellite (GLASS) remote sensing data products.

### 2.1.1 FAO Statistical Data

The global food production data of various countries from FAOSTAT (FAOSTAT) was used as the baseline data for the production allocation model. Specifically, the data consists of the production of four major crops (maize, wheat, rice and soybean), with countries as the statistical units. The data are measured in thousands of tonnes and the years covered by the data are from 2010 to 2020.



### 2.1.2 GAEZ+ 2015 Annual Crop Data

The 2015 GAEZ gridded crop production dataset and gridded crop harvested area dataset (Grogan et al., 2022) were used as training data to train the production spatial allocation model. These data are presented in a gridded format with a grid size of 10KM × 10KM. The pixel values in the production and harvested area datasets represent crop production and harvested area within each grid. Data were used for four types of crops, namely maize, wheat, rice and soybean.

### 2.1.3 CropWatch Crop Phenology Data

To better represent crop characteristics at different growth stages, we performed precise temporal window segmentation of additional data using phenology data. The crop phenology data (Zheng et al., 2016) was obtained from CropWatch Cloud (CropWatch Cloud, 2023), which is based on extensive field observations and scientific experiments. These data primarily provide information on the growing and harvesting periods of major crops across various countries globally, with a temporal resolution of 10 days.

### 2.1.4 CropWatch Agro-Ecological Zones

In this study, we used the agro-ecological zones (AEZs) data from CropWatch Cloud (CropWatch Cloud, 2023), which covers 228 agro-ecological zones in 45 countries around the world. These data are based on multiple factors such as climate, soil and topography in different parts of the world, and they comprehensively divide different agricultural ecological zones. These ecological zones represent regions with similar agricultural production conditions and crop planting patterns, and are of great value in understanding and predicting the global distribution of crop production. In this study, we used the divided ecological zones as the smallest modelling scale and built corresponding production spatial allocation models for each ecological zone. For countries or regions without subdivided agro-ecological zones, modelling was carried out using the country/region and other administrative units as homogeneous areas.

### 2.1.5 HWSD Soil Texture Data

In this study we used the soil texture data (Fischer et al., 2008) from HWSD. These data describe the texture of soils around the world, including the proportions of clay, sand and silt. Soil texture affects the ability of the soil to retain moisture and fertility, and therefore has a direct impact on crop growth and production. In our model, these soil texture data were used as a feature input to the model for training.

### 2.1.6 CropWatch Irrigated Land Distribution Data

In this study, we used global irrigated farmland distribution data (Wu et al., 2023a) from CropWatch Cloud as one kind input features. These data provide the types of irrigation across the world's cropland, including irrigated and rainfed types. Irrigation plays a key role in ensuring stable and high crop production, especially in arid and water-scarce regions. In our model, the irrigation type data (irrigated, rainfed and unknown) were uniquely coded and transformed into a three-dimensional feature for modelling.

### 2.1.7 Location Data

We used latitude and longitude data to represent the geographical location of each sample. Latitude and longitude data are crucial for capturing the influence of geographical location on crop production, such as solar radiation conditions at different latitudes and climatic zone characteristics at different longitudes. However, in model construction, latitude and longitude coordinates differ from Cartesian coordinates; Cartesian coordinates have smooth and uniform variations, while latitude and longitude, as polar coordinates, have uneven numerical changes and are not suitable for direct expression or measurement of positional or relational changes. Therefore, in this study we overcame this problem by converting latitude and longitude polar coordinates into coordinate features in three-dimensional Cartesian coordinates through geospatial encoding.



### 2.1.8 Terrain Data

In this study, we used global terrain data, including elevation and terrain variation coefficients.
Topographical factors affect climatic conditions and water flow, thereby influencing crop growth and production. In our model, topographic data serve as an essential environmental feature used to train the production distribution model. All topographic data have been standardised to reduce the influence of dimensions and to improve the generalisability of the model.

### 2.1.9 CropWatch Agroclimatic Indicator Data

We used the agroclimatic indicator data from CropWatch Cloud(CropWatch Cloud, 2023), including cumulative potential biomass (BIOMASS), cumulative precipitation (RAIN), photosynthetically active radiation (PAR) and average air temperature (TEMP). These indicators are time-series data, with BIOMSS available four times a year and RAIN, PAR and TEMP available 36 times a year. These data reflect the energy and moisture conditions of agricultural ecosystems. In
processing these data, we used crop phenology data to slice these agro-meteorological indicators into time windows and calculated the maximum, minimum, standard deviation and total sum within each time window as feature inputs to the model.

### 2.1.10 CropWatch Agronomic Indicator Data

In this study we used the agronomic indicator data from CropWatch Cloud(CropWatch Cloud,
2023), including the cropped arable land fraction (CALF) and the maximum vegetation condition index (VCIx). The CALF is the ratio of planted area to total cultivated area, calculated based on normalized difference vegetation index (NDVI). The VCIx is used to describe the historical level of vegetation condition during the observation period. A value of 0 indicates that the vegetation condition corresponds to the worst level of recent decades; 1 indicates that the vegetation condition corresponds to
the best level of recent decades; and a value greater than 1 indicates that the vegetation condition of the current observation period exceeds the historical optimum level. These indicators are time-series data with four periods per year. These indicators reflect crop growth conditions and area. In processing these

data, we used crop phenology data to divide these crop condition indicators into time windows that serve as feature inputs to the model.

### 150 2.1.11 GLASS Remote Sensing Data Products

In this study, we used GLASS remote sensing data products (Liang et al., 2013) including net primary productivity (NPP) and leaf area index (LAI). These data can comprehensively and accurately reflect the growth status of vegetation and the intensity of photosynthetic activity. Specifically, NPP is provided as annual data, while LAI is a time series data with a period every 9 days. In processing the

LAI data, we used crop phenology data to divide it into time windows and calculated the maximum, minimum, standard deviation and sum of the data within each time window, which served as feature inputs for the model.

### 2.2 Methods

The production process of the GGCP10 dataset consists of four main steps: harvest area estimation,

indicator data processing, data-driven model training and production calculation, as shown in Fig. 1.





**Figure 1. Flowchart for generating the GGCP10.**

## 2.2.1 Harvested Area Estimation

To calculate the crop production of a grid cell, its harvested area must first be determined. A simple method is to calculate the harvested area of each grid cell in the target year based on statistical data and gridded harvested area data from a reference year using proportional allocation (Grogan et al., 2022). However, this approach ignores changes in cropping conditions between different grid cells within a region.



Based on a multi-scale correlation analysis using GAEZ+ 2015 harvested area data and CropWatch
cropped area fraction data, we found that the gridded harvested area is significantly positively correlated
with the contemporaneous cropped area fraction. Therefore, to estimate the gridded harvested area more
accurately for the target year, we proposed a harvested area estimation method based on the change in
cropped area fraction. The main steps are as follows:

(1)Preparation of reference data

The gridded harvested area data $H_{ij}^{ref}$ from GAEZ+ 2015 data were used as a reference, where $i$ is
the grid cell index and $j$ is the crop type. The crop types contained in each grid and their proportion of
harvested area in each grid were then calculated.

(2)Calculation of the cropped area (CA)

For the reference year, derive the total cropped area $CA_{it}$ of each grid cell $i$ in season $t$ from the
cropped area fraction data, where $CA_{it}$ is the planted area of grid cell $i$ in season $t$. According to the
crop calendar information and the grid cell cropping fractions of crop $j$, divide $CA_{it}$ into different crop
types to obtain the cropped area $CA_{ij}^{ref}$ of each grid cell $i$ for crop $j$ in the reference year. Similarly, first
the total area data $CA_{it}$ of each grid cell $i$ in different seasons for the reference year are obtained and
then the area $CA_{ij}^{tar}$ of each grid cell $i$ for crop $j$ is calculated.

(3)Calculate the rate of change of the cropped area

This step is carried out separately for each crop $j$. For each grid cell $i$, compare its area $CA_{ij}^{tar}$ in the

target year with $CA_{ij}^{ref}$ in the reference year to derive the area change ratio $r_{ij} = \frac{CA_{ij}^{tar}}{CA_{ij}^{ref}}$. The total rate of

change in the area $r_j = \frac{\sum_j CA_{ij}^{tar}}{\sum_j CA_{ij}^{ref}}$ is then calculated for crop $j$ within the administrative unit.

(4) Calculation of proportions of harvested area

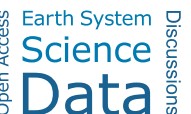

This step is also carried out separately for each crop $j$. From the reference data, the proportion of the harvested area of grid cell $i$ in the total harvested area of the administrative unit for crop $j$ can be obtained, denoted as $w_{ij}^{ref} = \dfrac{H_{ij}^{ref}}{\sum\limits_{j} H_{ij}^{ref}}$.

    For crop $j$, the percentage of harvested area $w_{ij}^{ref}$ of grid cell $i$ in the target year shall be calculated according to the following formula, based on the percentage change in area $r_{ij}$ of each grid cell $i$, the

total percentage change in area $r_j$ of crop $j$ in the administrative unit and the percentage of harvested area $w_{ij}^{ref}$ of grid cell i in the reference year:

$$w_{ij}^{tar} = w_{ij}^{ref} \times \frac{r_{ij}}{r_j}$$

    (5) Calculate the harvested area

    Obtain the total harvested area $H_j^{tar}$ of crop $j$ within the administrative unit in the target year from

the FAO data. Then use the proportion of harvested area $w_{ij}^{tar}$ of each grid cell $i$ to calculate its harvested area $H_{ij}^{tar} = H_j^{tar} \times w_{ij}^{tar}$. Finally, consistency processing is performed to ensure that the sum of the harvested areas of all grid cells within each administrative unit matches the statistical data after calculation.

    Through this approach, the estimation of the harvested area considers the changes in the cropped

area between different grid cells within a region, reducing the deviations caused by simple proportional allocation and thus allowing a more accurate estimation of the spatial distribution of the harvested area. In addition, to improve the reliability of the results, we detect outliers in the cropped area fraction data and apply interpolation to smooth the outliers, eliminating the effects of spatial anomalies commonly caused by clouds, shadows and snow cover in remote sensing imagery. This increases the robustness of

the harvested area estimate and avoids the overall bias introduced by a small number of contaminated pixels.



### 2.2.2 Indicator Data Processing

Indicator data processing involves data clipping based on crop phenology and feature extraction from the data. First, the time series data are clipped by crop phenology to obtain the data corresponding
to the crop growth period. Features are then extracted from each type of data to form the feature vector for each crop, which is used in subsequent model training. All the features used for model training are listed in Tab 1.

**Table 1. Input features**

| Feature name | Feature type | Images per year | Dimensions |
|---|---|---|---|
| Harvested area | Annual data | 1 | 1 |
| Maximum vegetation condition index (VCIx) | Time series | 4 | 4 |
| Cropped arable land fraction (CALF) | Time series | 4 | 4 |
| Cumulative potential biomass (BIOMSS) | Time series | 4 | 4 |
| Cumulative precipitation (RAIN) | Time series | 36 | 4 |
| Photosynthetically active radiation (PAR) | Time series | 36 | 4 |
| Average air temperature (TEMP) | Time series | 36 | 4 |
| Net primary productivity (NPP) | Annual data | 1 | 1 |
| Leaf area index (LAI) | Time series | 46 | 4 |
| Location (gx, gy, gz) | numerical value | - | 3 |
| Terrain (elevation and terrain variation coefficients) | numerical value | 2 | 2 |
| Soil (clay, sand and silt) | Category | 3 | 3 |
| Irrigation Type (irrigated, rainfed and unknown) | Category | 1 | 3 |
| Total | - | 174 | 41 |





### 2.2.3 Data-Driven Model Training

There is a close correlation between crop production and the corresponding harvested area (HA) and multi-source indicators for each grid cell, and these correlations are largely consistent within local regions. This principle facilitates the development of data-driven models for allocating gridded production (P) within each crop and agro-ecological zone (AEZ) as follows:

$$P_i^j = f\left(HA_i^j, XI_i^j\right)$$

Where the variables $P_i^j$, $HA_i^j$ and $XI_i^j$ respectively represent production, harvested area, and various multi-source indicators in grid cell $i$ of crop type $j$. The $f$ is a machine learning model customized for each AEZ and crop type to capture their unique relationship between production, harvested area and indicators.

For each specific crop type, the data-driven models were independently built for each AEZ based on its geographical subdivision. That is, one model was trained for each crop in each AEZ. Specifically, two steps were taken:

**(1) Optimal model selection**

Within an agro-ecological zone, due to the complexity and diversity of influencing factors, the predictive performance and parameter optimisation of different machine learning models may vary. Therefore, for AEZ, model selection and parameter optimisation are first carried out. The specific operation is as follows: the data from the reference year are divided into training and test sets, where the training set is mainly used to optimise model parameters, and the test set is used to evaluate model prediction performance, thereby supporting our final model selection.

We select three widely used machine learning models, Random Forest (Breiman, 2001), XGBoost (Chen and Guestrin, 2016) and CatBoost (Prokhorenkova et al., 2019), as candidate models, and use grid search and cross-validation methods to optimise the parameters of each model. Then, based on the optimised parameters, we train each model on the training set and evaluate its predictive ability on the test set, with the evaluation metric chosen as $R^2$. Finally, the model with the highest $R^2$ value is selected as the optimal model for that ecological zone.

**(2) Building predictive models**

In the optimal model selection stage, 90% of the reference year data are used as the training set. Given the limited data resources and their importance for prediction, we prefer to make full use of all the reference year data. Therefore, we combine all the reference year data with the optimal parameters

obtained in the previous step to train the models in order to improve the model stability and prediction accuracy, thus obtaining production spatial allocation models for each AEZ.

### 2.2.4 Production Calculation

For each AEZ and each crop type, based on its spatial production allocation model, the feature data of this zone are imported to obtain the predicted production of each grid as the initial grid production,

denoted as $P_{ij}^{init}$.

However, while our data-driven models and harvested area estimations provide a robust foundation, it's essential to recognize that model-derived predictions might not always perfectly align with established agricultural statistics. Such deviations can emerge from various factors, including model limitations, data anomalies, or unforeseen agricultural events.

It is important to recognize that, despite the robust foundation provided by our data-driven models and harvested area estimates, the model-derived predictions may not always be perfectly consistent with established agricultural statistics. Such discrepancies may arise due to various factors such as model limitations, data anomalies or unexpected agricultural events. Therefore, data consistency processing is necessary.

The reconciliation process involves recalibrating the initial grid production $P_{ij}^{init}$ based on the FAO's statistics. This recalibration ensures that the aggregated production figures across all grid cells within an administrative unit align with the FAO's reported data. The formula below is meticulously applied to achieve this alignment:

$$P_{ij} = \frac{P_j}{\sum_i P_{ij}^{init}} \times P_{ij}^{init}$$

Where $i$ represents a grid, and $j$ represents a crop. $P_j$ denotes the crop production statistics at the administrative unit level, and $P_{ij}$ denotes final grid production.



The alignment at the administrative unit level ensures that our dataset is a dependable tool for granular and macro-level agricultural analyses.

# 3 Results and Discussion

## 3.1 Spatial Features of Crop Production in GGCP10

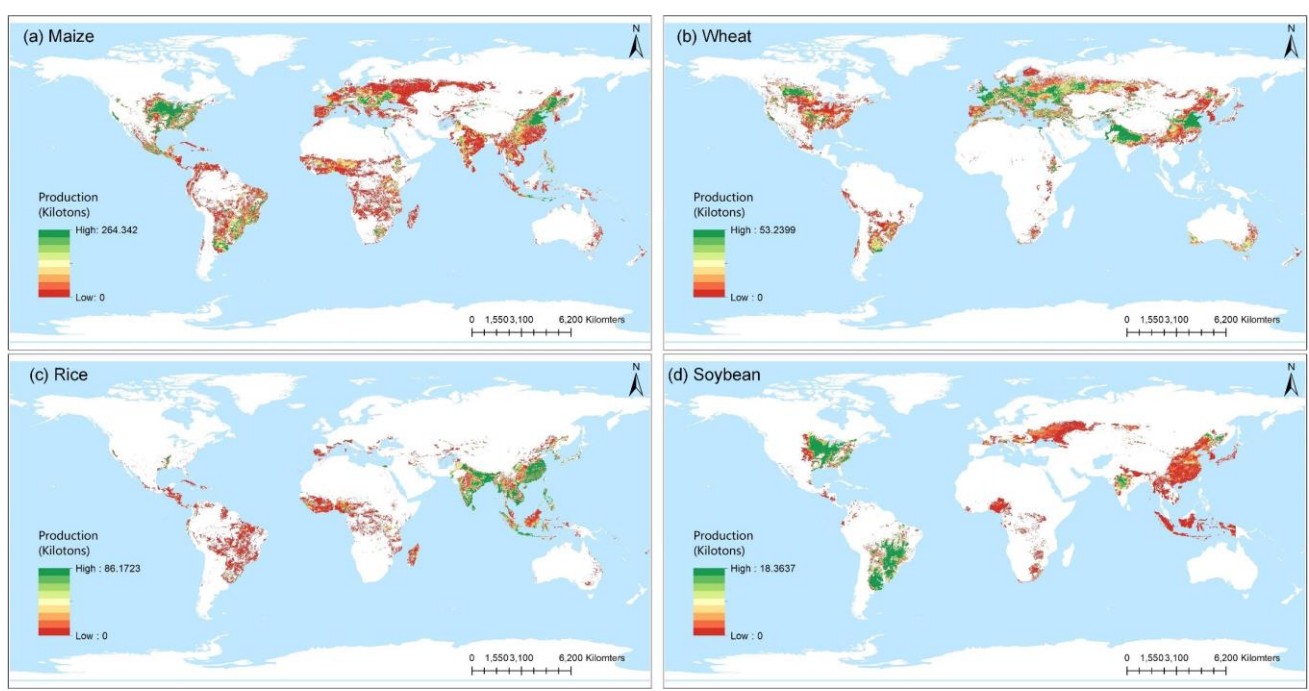

**Figure 2. Production distribution of GGCP10 in 2020: (a) Maize; (b) Wheat; (c) Rice; (d) Soybean.**

The production distribution patterns in 2020 for the four crops are illustrated in Fig 2. In the case of maize, regions with higher production are located in the Corn Belt of the United States, southern
Brazil, the wet Pampas of Argentina, the northwestern Black Sea regions of Ukraine and Romania, and northeastern and northern China. Regarding wheat, regions with higher production include southern Canada, Argentina, Europe, the Nile River Delta, North China, and northern India. For rice, regions such as South China, Southeast Asia, and South Asia have a higher production. As for soybeans, grid cells with higher production are primarily concentrated in the eastern Great Plains of the United States,

southern Brazil, and northern Argentina, but there are also scattered high production areas in Northeast China and central India.

The spatial distributions of production shown in the dataset are consistent with the general knowledge for these four crops. This validates GGCP10's ability to provide precise insights into global crop production patterns. The hotspots on a continental scale and regional clusters demonstrate how
cropping systems and agro-climatic suitability affect crop distribution globally. Additionally, the spatial crop production patterns are consistent with expectations, showcasing how useful the dataset is for agricultural studies utilizing its exceptional spatiotemporal resolution and range.

Analyzing the spatial production maps can reveal crop expansion fronts and production variabilities within major breadbaskets. Comparing distributions across years may uncover geographic
shifts in response to climate or policy changes. The value of GGCP10 lies in enabling spatial research on crop production from local to global scales. Such analysis is particularly beneficial for investigating how crop production has changed over time in response to various factors. Its regular gridding and wall-to-wall coverage facilitate flexible geospatial analysis. In summary, the dataset affords more precise production information compared to national statistics, thereby triggering novel research avenues in
agriculture science and food security evaluations.

### 3.2 Model Performance Evaluation

### 3.2.1 Comparison of Different Models

To visualize the performance of the three models (XGBoost, CatBoost, and RF) in cross-validation (Fig. 3), we employed the Gaussian kernel probability density plot of $R^2$. $R^2$ is a measure of model
prediction accuracy, with a value closer to 1 indicating higher accuracy. The horizontal coordinate represents the $R^2$ value, the vertical coordinate shows the Gaussian kernel probability density of $R^2$, and the graph depicts the distribution of $R^2$ values across all models. By using this approach, we can evaluate and compare the overall prediction accuracy of the models.





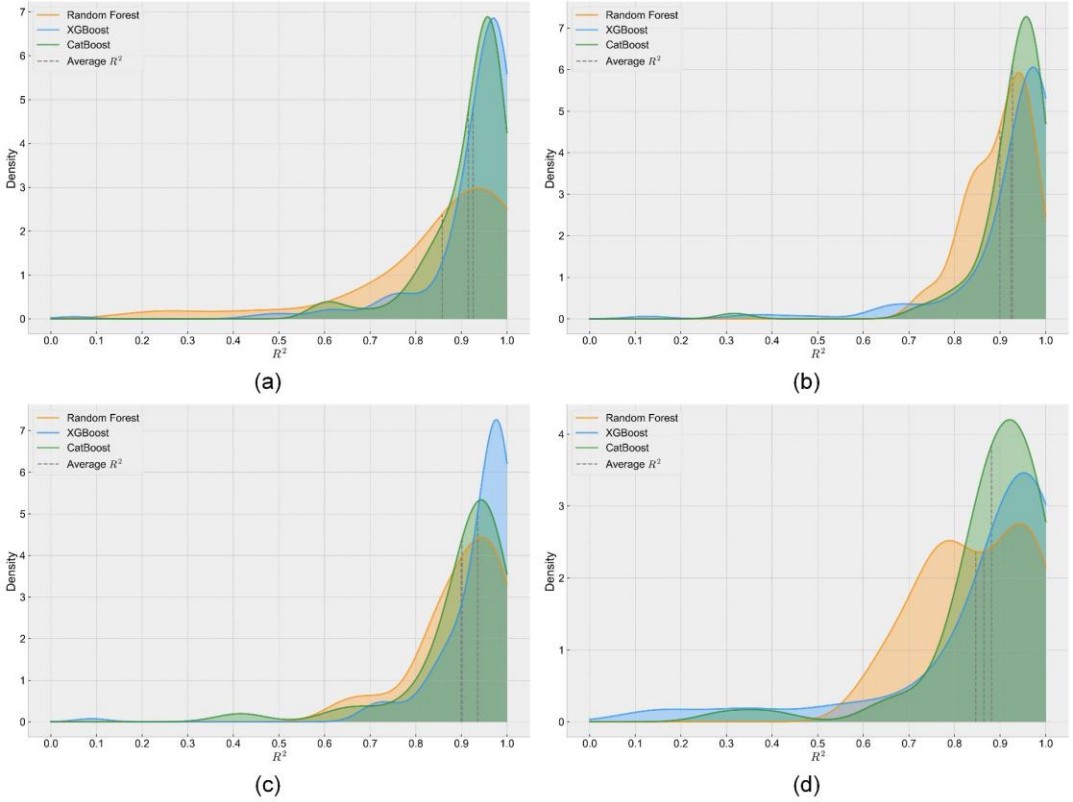

**Figure 3. Gaussian kernel probability density of $R^2$ for models: (a) maize; (b) wheat; (c) rice; (d) soybean**

We developed a total of 303 regional models for maize, with 199 XGBoost models, 79 CatBoost models, and 25 RF models (Fig. 3a). The XGBoost models had an average $R^2$ of 0.928 with a primary distribution between 0.90 to 1.00, which shows high prediction accuracy. The CatBoost models had an average $R^2$ of 0.915, with most values distributed between 0.86 to 1.00. The RF models had an average $R^2$ of 0.858, with a range between 0.72 to 1.00.

We trained 237 regional models for wheat, including 138 XGBoost models, 82 CatBoost models, and 17 RF models (Fig. 3b). Out of the three models, CatBoost achieved the highest $R^2$ values, primarily distributed between 0.92-1.00, with an average of 0.927. The XGBoost models exhibited $R^2$ values mainly distributed between 0.87-1.00, with an average of 0.924. The RF models demonstrated $R^2$ values mainly distributed between 0.75-1.00, with an average of 0.900.





Out of 202 rice models (Fig. 3c), XGBoost, CatBoost and RF models accounted for 145, 37 and 20 models, respectively. The $R^2$ values for XGBoost were primarily distributed between 0.90-1.00 with an average of 0.936. The $R^2$ values for CatBoost were mainly distributed between 0.85-1.00, with an average of 0.901. The $R^2$ values for RF models were primarily distributed between 0.80-1.00 with an average of 0.899.

Out of the 155 soybean models (Fig. 3d), 84 were XGBoost, 54 were CatBoost, and 17 were RF. XGBoost had an average $R^2$ value of 0.863, with most of the values distributed between 0.88-1.00; CatBoost had an average $R^2$ value of 0.880, with most of the values distributed between 0.80-1.00; and RF had an average $R^2$ value of 0.845, with most of the values distributed between 0.70-1.00.

The probability density plots indicate that for all four crops, the $R^2$ values of XGBoost, CatBoost, and RF are high, with narrow ranges, demonstrating good model accuracy and stability. For all four crops, the number of XGBoost models is significantly higher than that of CatBoost or RF, suggesting that XGBoost has the best model performance in most regions, which is consistent with the conclusions from our previous study(Li et al., 2023). The number of models for each crop reflects the global area of that crop, as we only trained models in regions where we had corresponding data. For instance, maize has the highest number of models owing to its widespread distribution. These results demonstrate the potential of selecting optimal models and parameters adaptively to train relatively accurate models for various crops and regions.

### 3.2.2 Evaluation of Model Performance in Different Regions

The purpose of this section is to evaluate and compare the performance of models in different geographical regions. The models of various regions have been grouped according to their continents, and their $R^2$ distributions for the four crops in different regions at the continental scale are depicted in the figure below (Fig. 4). Violin plots have been utilized to present the $R^2$ distributions of models in different regions within each continent. These plots also display the maximum, minimum, median, and quartile ranges.



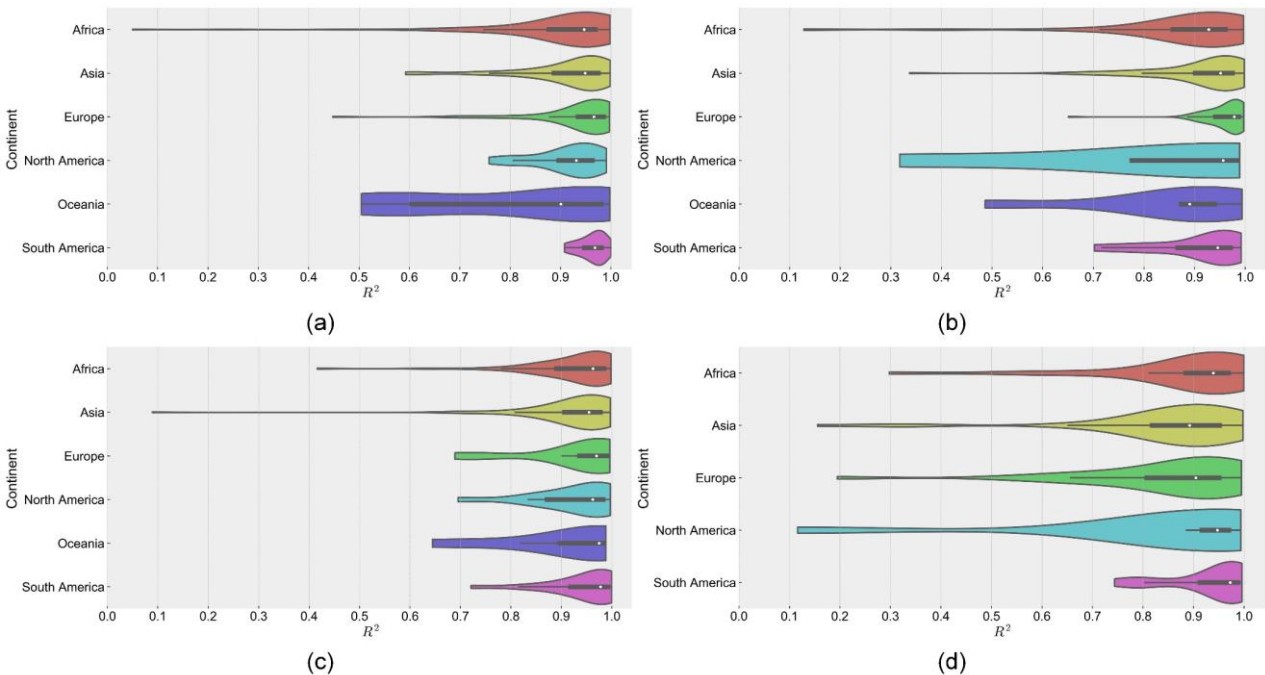

**Figure 4. Evaluation of model performance in different regions: (a) maize; (b) wheat; (c) rice; (d) soybean**

The data suggest that most models perform well, with only a few showing lower accuracy. Regarding model performance characteristics, the following observations have been made for each continent:

In Africa, the fluctuation range is substantial for wheat models, with an average $R^2$ of 0.878. In contrast, the accuracy of the remaining three crops' models is relatively consistent.

In Asia, the soybean models depict higher variation range, averaging an $R^2$ of 0.848. Conversely, rice models are the most accurate amongst the four crops with an $R^2$ of 0.922.

In Europe, crop models showcase relatively lower variation ranges except for soybean. Wheat models demonstrate an $R^2$ of up to 0.960, while soybean models have an average $R^2$ of 0.843.

In North America, maize and rice models are more stable, whereas wheat and soybean models have comparably wider variation ranges. Rice models have the highest average $R^2$ of 0.926, while wheat models have the lowest average $R^2$ of 0.805.

In South America, models for all four crops in different regions show high accuracy with an average $R^2$ greater than 0.910 and small variation ranges.

In Oceania, maize models show a wider variation range, which may be due to smaller maize planting areas in this region, resulting in inadequate samples.

The data shows that model performance for different crops varies across regions, which is likely
due to factors such as climate, soil conditions, and cropping patterns specific to each region. Overall, the models for different crops in various regions demonstrate relatively high accuracy, further reinforcing the reliability of the trained models in this study.

**3.3 Feature Importance Analysis**

In this section, we look more closely at the importance of different features in the crop production
prediction models for different crops, and try to explain this importance in combination with crop physiology.

First, we calculate the importance of each feature in each regional model during training, and calculate the average value across all models for each feature; meanwhile, we also summarise the importance of subsets within each feature class to obtain the overall feature importance. Since harvested
area was provided as a prior and had the highest importance universally, we focused our analysis only on the remaining features aside from harvested area.





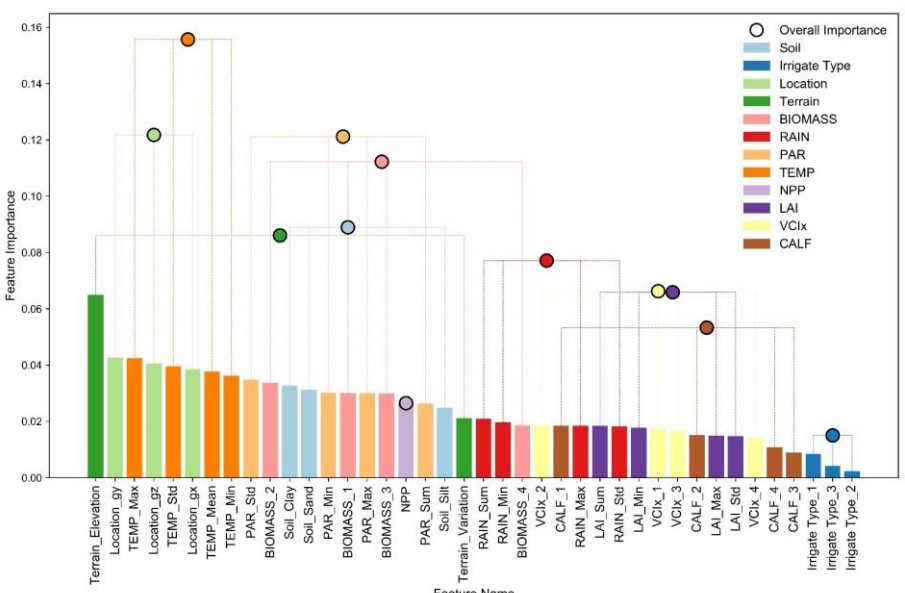

**Figure 5. Feature importance scores of maize.**

For maize (Fig. 5), the TEMP class (including maximum, minimum, mean, standard deviation of average air temperature during the growing season) has the highest importance. This corresponds to the temperature sensitivity of maize physiology(Hsiao et al., 2019; Feng et al., 2019; Zhang et al., 2022), especially at key growth stages. The second is Location, showing the significant effect of geographical location on maize yield. The third is PAR (photosynthetically active radiation), which is explained by the high light requirements of maize, with PAR intensity having a direct effect on photosynthesis and yield. Among the individual characteristics, altitude is the most important, probably due to the direct effect of altitude on temperature and climatic conditions, and thus on maize growth.



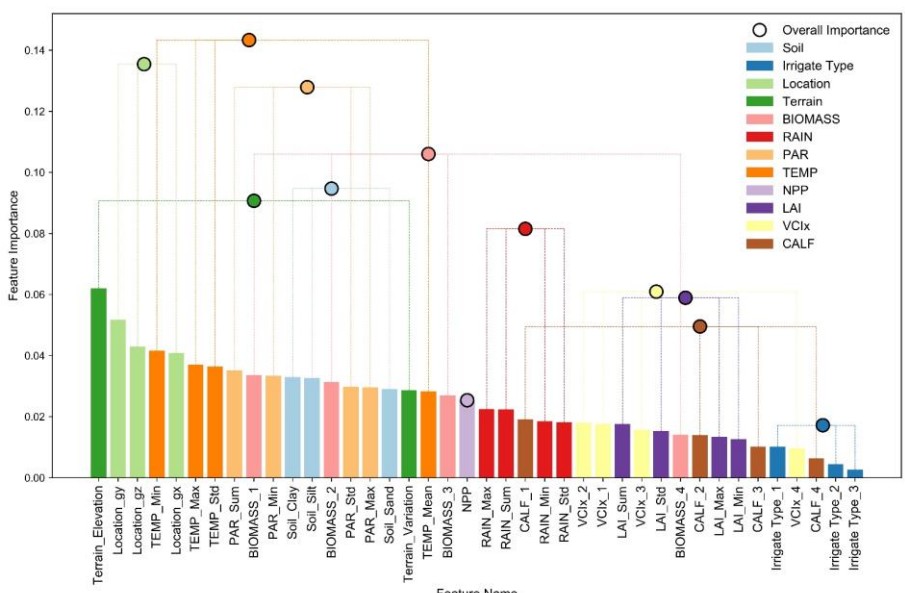

**Figure 6. Feature importance scores of wheat.**

For wheat (Fig. 6), the three most important characteristics are TEMP, Location and PAR. Wheat

growth is sensitive to temperature, especially at the sowing and heading stages, resulting in specific

temperature requirements(Perdomo et al., 2016). TEMP is therefore the most important. Meanwhile, the

high adaptability of wheat to soils and environments makes geographical location factors particularly

important. The effect of sunlight on wheat is also not negligible, leading to a high importance of PAR.

Similar to maize, altitude is the most important of the individual characteristics.






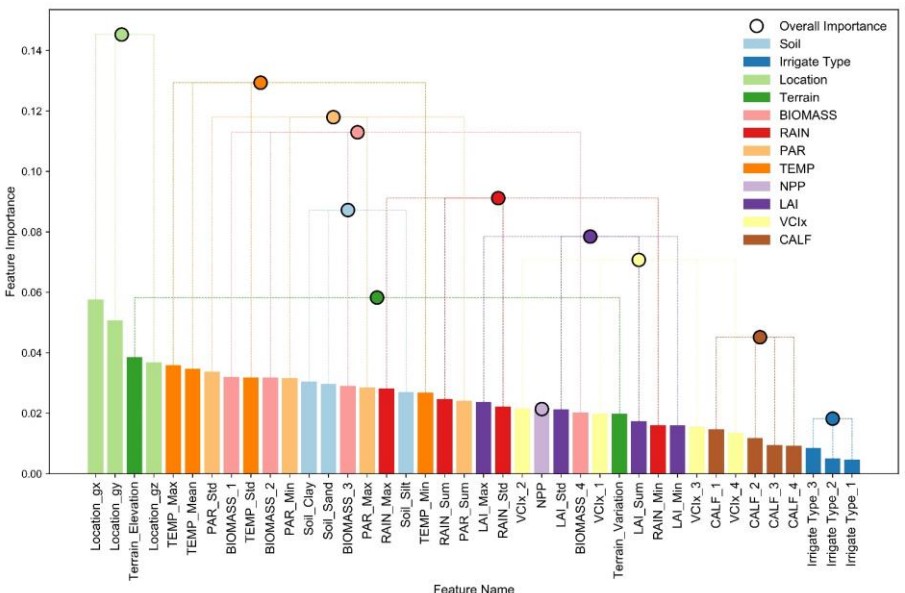

**Figure 7. Feature importance scores of rice.**

For rice (Fig. 7), the order of importance for the top three characteristics is Location, TEMP and PAR, possibly because these characteristics are directly related to the growing environment, temperature and photosynthesis of the crop(Su et al., 2023; Perdomo et al., 2016). In addition, 400 BIOMASS and RAIN also have relatively high importance, probably related to biomass accumulation and water requirements of rice(Yan et al., 2022). Among the individual characteristics, the grid coordinates (gx and gy) are more important.





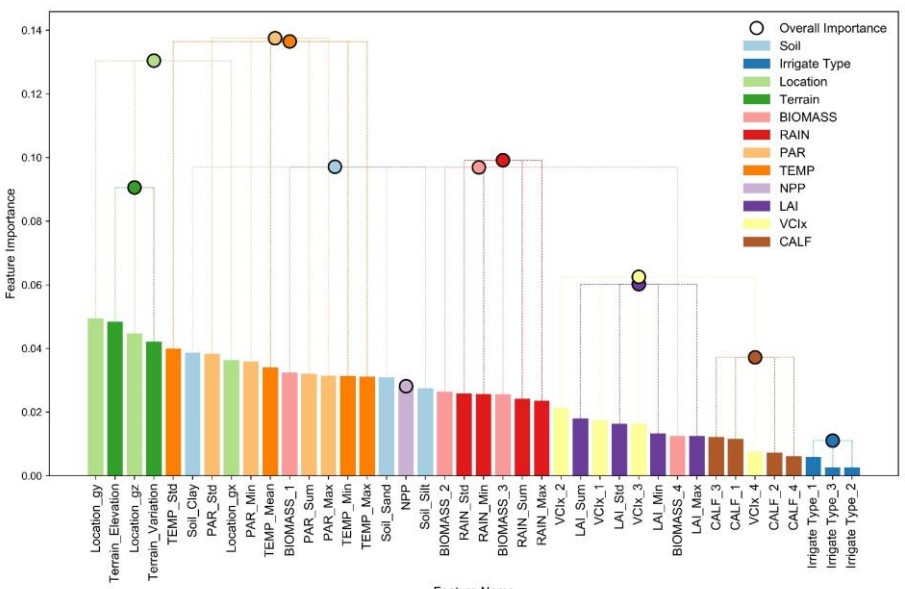

**Figure 8. Feature importance scores of soybean.**

For soybean (Fig. 8), PAR and TEMP have the highest importance, possibly due to the significant effects of variations in radiation and temperature during growth on soybean yield(Lin et al., 2023). Location is also highly significant. In addition, RAIN, Soil, Terrain and BIOMASS also have relatively high importance, possibly related to the dependence of soybean on soil, rainfall, topography and biomass accumulation. Among the individual features, gy was the most important, followed by altitude,

probably because latitude (gy) and altitude determine the climate and geography in which soybean grows(Li et al., 2023).

        In summary, TEMP and PAR show high importance for all four crops, highlighting the fundamental effects of temperature and radiation on crop growth(Perdomo et al., 2016). Similarly, Location also shows high importance for all crops, highlighting the critical role of geographical location

in determining crop yield(Li et al., 2023). In contrast, Irrigate Type has a lower importance for all crops.

## 3.4 Comparing with Existing Datasets

        To assess the reliability and accuracy of the proposed GGCP10 dataset, we evaluated it against reference datasets, including gridded, statistical, and survey data. Although the years and regional



coverage of these reference datasets may not entirely encompass the GGCP10 dataset, they still serve as
benchmarks to evaluate its reliability and accuracy from various perspectives, thereby providing
significant comparative value.

For comparison, we used two gridded datasets - SPAM 2010 and AsiaRiceYield4KM - as
reference datasets. We selected data from the same years across these datasets to conduct consistency
analysis with our dataset.

We obtained statistical data on crop production from the DES for major crop producing states in
India. These data are relatively reliable at the state level. We used this data as the baseline to assess the
accuracy of our dataset in India's state administrative units.

The survey data used were obtained from the USDA in the United States and were stratified by
county. Despite its low reliability, comparing the derived survey data with them can still offer valuable
insights into the dataset's reliability.

### 3.4.1 Comparison with SPAM 2010

SPAM 2010(Yu et al., 2020) is a commonly used global agricultural production dataset. To
appraise the dependability of GGCP10, we chose four crops (maize, wheat, rice, and soybean) from
SPAM 2010 for comparative analysis. Given that the unit of measurement in SPAM 2010 is tonnes and
GGCP10 is expressed in kilotons (kt), we converted the SPAM 2010 data into kilotons for a more
consistent comparison. Scatter plots and kernel density estimation plots were generated to visualize and
evaluate the coherence between the two datasets (Fig. 9).

Earth System Discussions
Open Access Science
Data

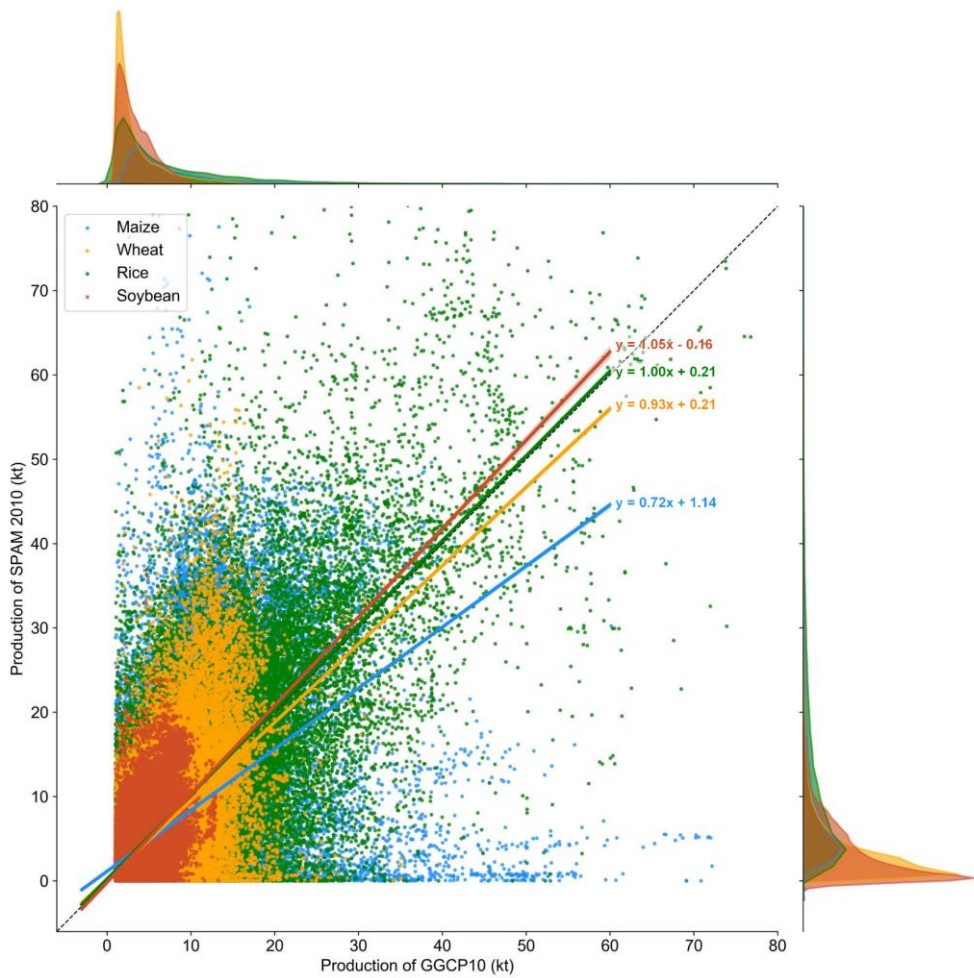

**Figure 9. Scatter plot and marginal distribution: comparing GGCP10 with SPAM 2010.**

To quantitatively assess the agreement, we conducted linear regression analysis on the two datasets. The results uncovered that the regression slopes for wheat, rice, and soybean were almost identical to 1, with $R^2$ values of 0.39, 0.55, and 0.35, and RMSE values of 3.61 kilotons, 7.33 kilotons, and 2.87 kilotons, respectively. This denotes a high level of consistency between the datasets. Conversely, the maize slope was marginally lower at 0.72, but still acceptable, with an $R^2$ of 0.28 and

an RMSE of 6.35 kilotons.

        The kernel density estimation plots show the distribution of crop production quantities for each crop. The GGCP10 and SPAM 2010 pixel values were mainly concentrated in the 0-10 kilotons range,





indicating good alignment. Rice had the most significant distribution divergence, with the GGCP10 peak value corresponding to lower pixel quantities than SPAM 2010.

### 450 3.4.2 Comparison with AsiaRiceYield4km

The AsiaRiceYield4km(Wu et al., 2023b) dataset provides a high-resolution (4KM) seasonal grid of rice yields in Asia, spanning from 1995 to 2015, and covers single, double, and triple-season rice. In the interest of harmonizing the evaluation metrics and ensuring consistency with our GGCP10 dataset, several adjustments were made to the AsiaRiceYield4km dataset.

Due to the unavailability of seasonal harvested area data in AsiaRiceYield4km, the comparison was constrained to single-season rice areas, which constitute 56.5% of the total AsiaRiceYield4km extent. To align with the spatial resolution of GGCP10, the AsiaRiceYield4km dataset was resampled to a 10km grid. In the development of the GGCP10 dataset, we also generated corresponding harvested area data, allowing us to calculate total production values for AsiaRiceYield4km based on these areas. 460 These recalculated total production data served as the basis for consistency evaluation with the GGCP10 dataset.

For the overlapping years from 2010 to 2015, scatter density plots (Fig. 10) were used to assess the consistency of gridded production data between GGCP10 and AsiaRiceYield4km.



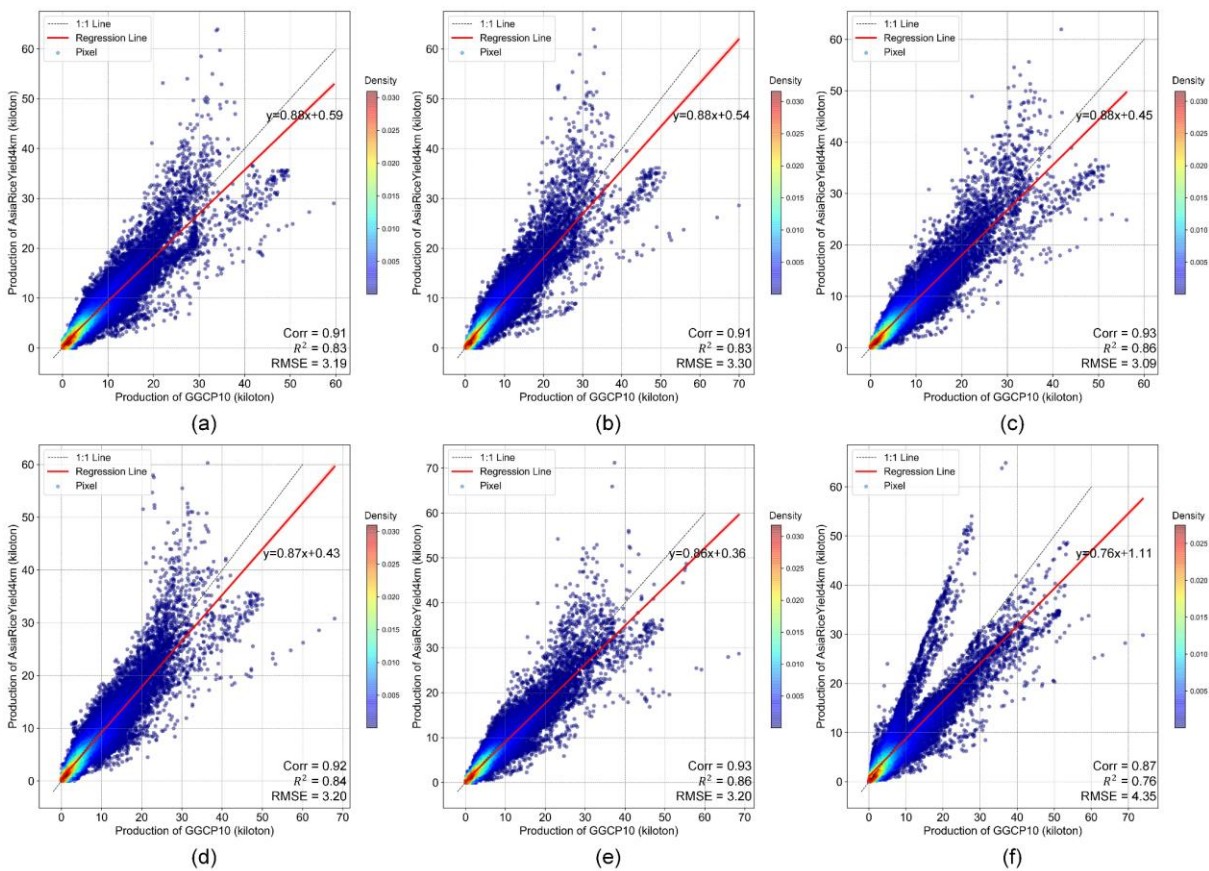

**Figure 10. Scatter density plots: (a) 2010; (b) 2011; (c) 2012; (d) 2013; (e) 2014; (f) 2015.**

The data reveal a strong positive correlation between GGCP10 and AsiaRiceYield4km for the years 2010-2014. Data points closely align along the 1:1 line, reinforcing that GGCP10 accurately captures the data distribution patterns present in AsiaRiceYield4km. Acceptable error rates are indicated by RMSEs ranging from 3.09 to 3.30 kilotons per 10km grid. The $R^2$ values range from 0.83 to 0.86, and correlation coefficients are between 0.91 and 0.93. Notably, the year 2015 exhibits a marginally lower slope and $R^2$, yet remains within acceptable limits. It should be highlighted that for all the years examined, the slope of the fitted line is less than 1, suggesting that GGCP10 tends to overestimate production when compared to AsiaRiceYield4km.

In summary, GGCP10 exhibits a strong degree of consistency with AsiaRiceYield4km in terms of single-season rice production grids, although localized discrepancies do warrant further investigation.



### 3.4.3 Comparison with Statistical Data from India DES

To further validate the effectiveness of the proposed GGCP10, we collected maize, wheat, rice and soybean production data from 2010 to 2020 in the major producing states of India from website of Directorate of Economics and Statistics (DES)(DES, 2023) from department of agriculture as a
reference. To increase the representativeness of the sample, multi-year data were aggregated for each crop due to potential random effects in single-year data.

We used correlation coefficients, RMSE and $R^2$ to quantify the consistency between GGCP10 and the Indian DES statistics, supplemented by scatter plots, with the results shown in Fig 11.

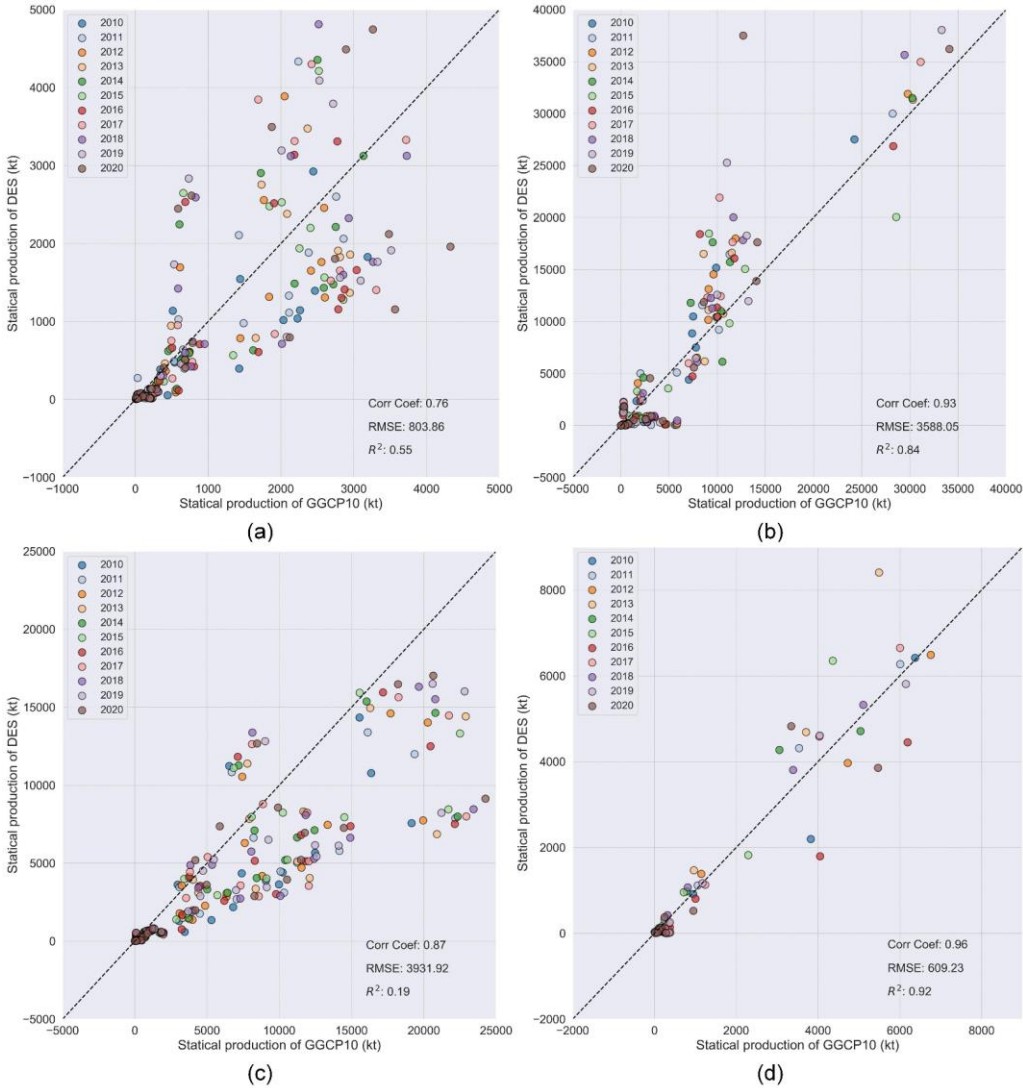

**Figure 11. Scatter plots of four crops: (a) maize; (b) wheat; (c) rice; (d) soybean**

The results show that although there are subtle differences in the consistency performance for each crop, GGCP10 shows significant agreement with the statistical data.

In particular, although maize shows some underestimation and overestimation in regions with higher total production, its correlation coefficient reaches 0.76 and $R^2$ is 0.55, indicating that GGCP10 remains reasonably consistent with the statistics to some extent. For wheat and soybean, the correlation coefficients exceed 0.93 and $R^2$ exceeds 0.84, indicating a higher consistency between these crops in

our dataset and the DES statistics. For rice, the correlation coefficient is 0.87, but the $R^2$ is only 0.19. The scatter plots show that GGCP10 has a systemic overestimation compared to the DES statistics. Overall, our dataset shows remarkable consistency compared to the Indian DES statistics, especially for
crops like soybean, wheat and maize where the consistency is more significant.

### 3.4.4 Comparison with USDA Data

We selected the United States Department of Agriculture (USDA) "SURVEY" data(USDA, 2023) as validation data (county level). This type of data is obtained from sample surveys rather than general statistics. Although they may be less comprehensive than CENSUS data, survey data are often more
flexible, more targeted and cover longer time periods. Despite the sampling errors inherent in their sample-based nature, they remain a very valuable reference in the absence of more comprehensive data. Therefore, although the use of survey data as validation data has limitations, it remains very important for our research.

To assess the consistency between GGCP10 and the USDA data, we used kernel density estimation
plot, as shown below. This type of plot shows the joint distribution of two variables on a two-dimensional plane, allowing us to visually identify data distribution densities. Darker coloured regions represent areas where data points are more concentrated, i.e. these points occur more frequently.




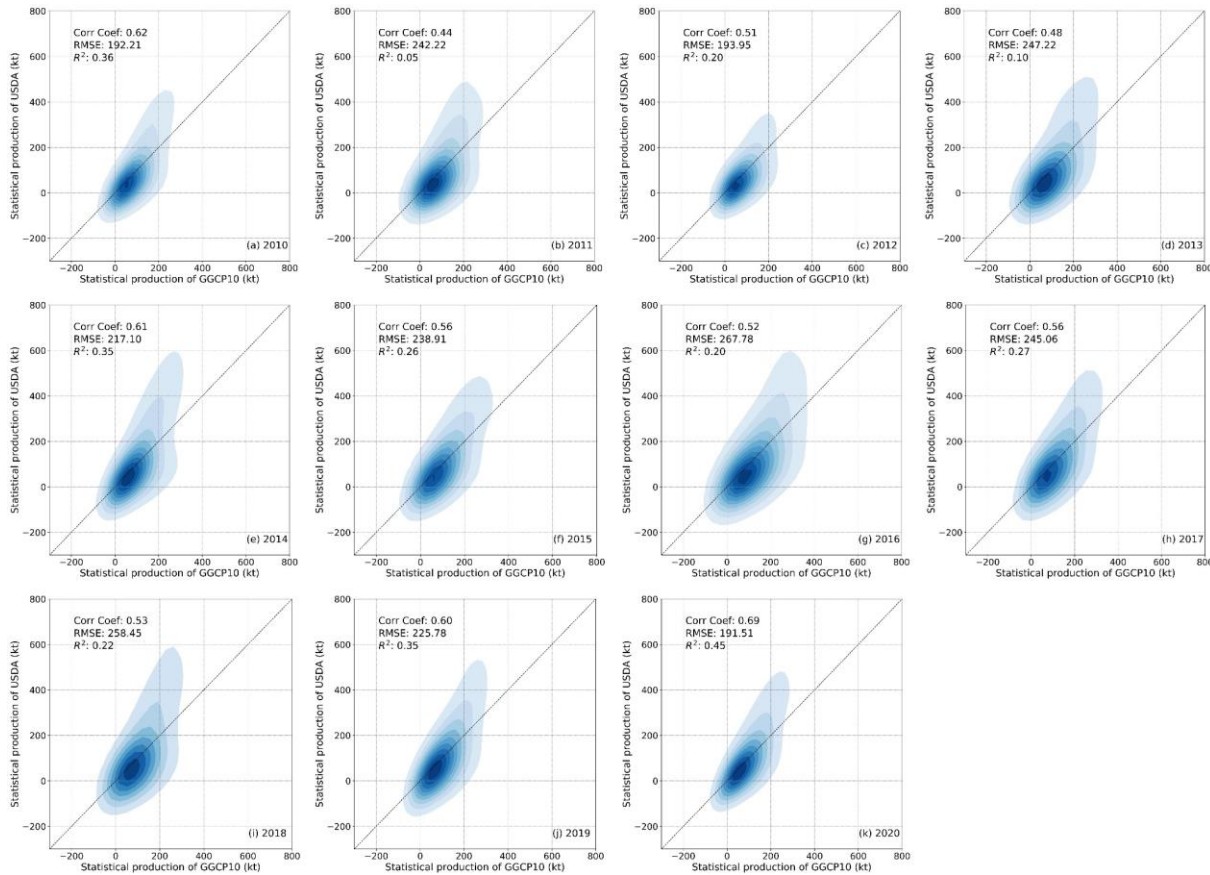

**Figure 12. Kernel density estimation plots of the two datasets of maize from 2010 to 2020.**

For maize (Fig. 12), the consistency between GGCP10 and the USDA data varies from year to year. The consistency is lowest in 2011 and 2013, while it is relatively higher in 2010, 2014, 2019 and 2020. Specifically, $R^2$ reaches a maximum of 0.45 in 2020, with an RMSE of 191.5 thousand tonnes and a correlation coefficient of 0.69, indicating relatively high consistency. Overall, however, GGCP10 tends to overestimate production in low-production areas and underestimate production in high-

production areas compared to the USDA survey data.



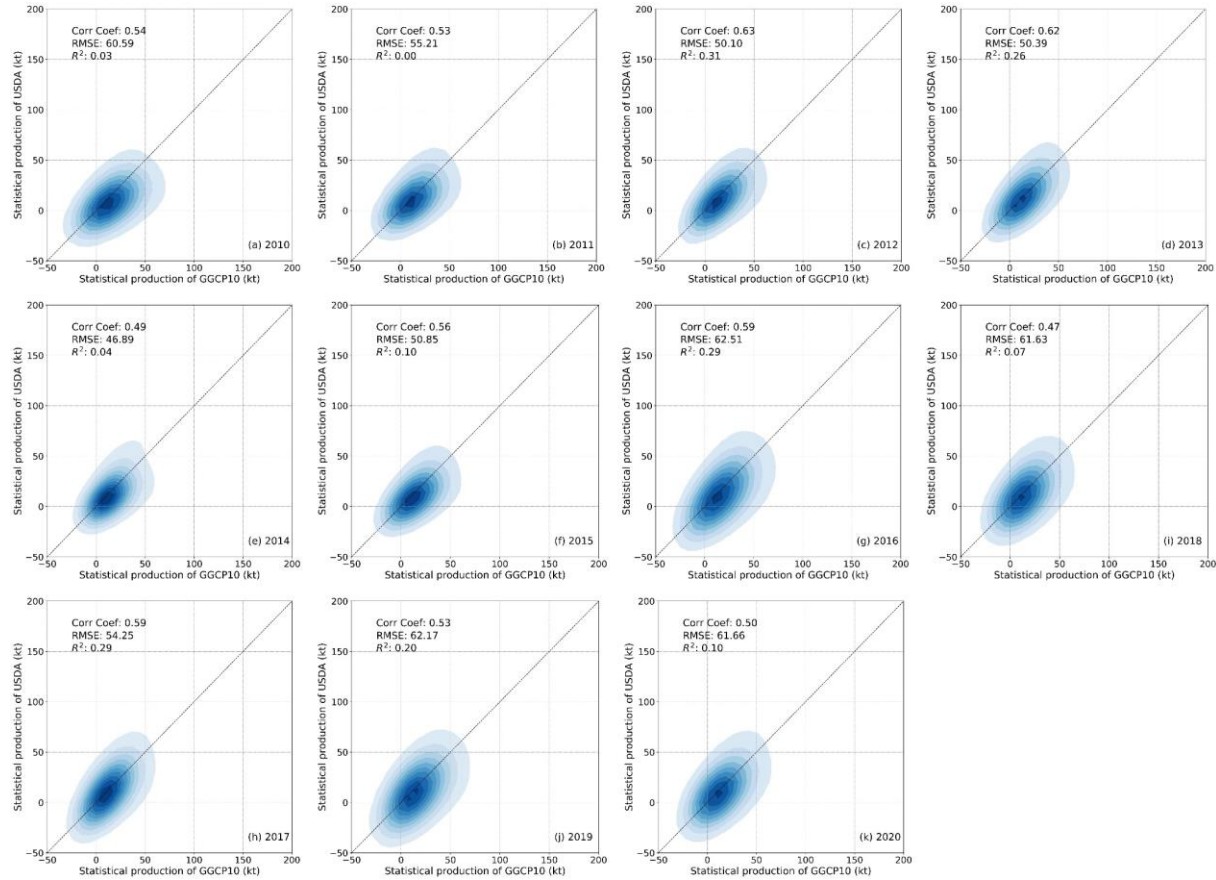

**Figure 13. Kernel density estimation plots of the two datasets of wheat from 2010 to 2020.**

For wheat (Fig. 13), although the $R^2$ values are relatively low, the lower RMSE and higher correlation coefficients indicate significant correlations between GGCP10 and the USDA data. Overall, the distributions of the data mainly follow the 1:1 line, and despite the lower consistency compared to other crops, the correlations remain generally high.

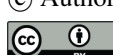



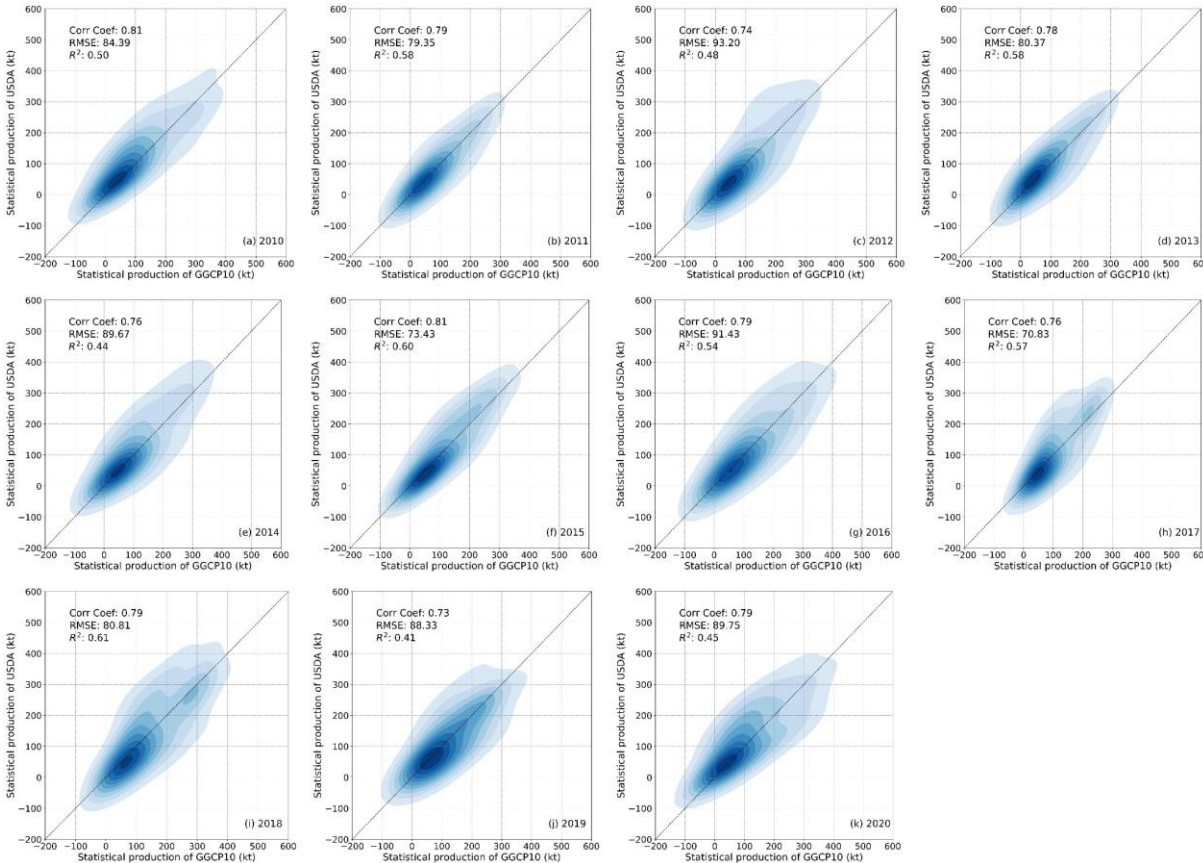

**Figure 14. Kernel density estimation plots of the two datasets of rice from 2010 to 2020.**

For rice (Fig. 14), GGCP10 shows no significant underestimation or overestimation in any of the 11 years when compared to the USDA data, demonstrating extremely high consistency. Over the 11 years, $R^2$ exceeds 0.50 for seven years, demonstrating the high correlation between the datasets. The lowest correlation coefficients in all years are 0.73 in 2012 and 2019, further demonstrating the consistency between our dataset and the USDA data in overall trends.





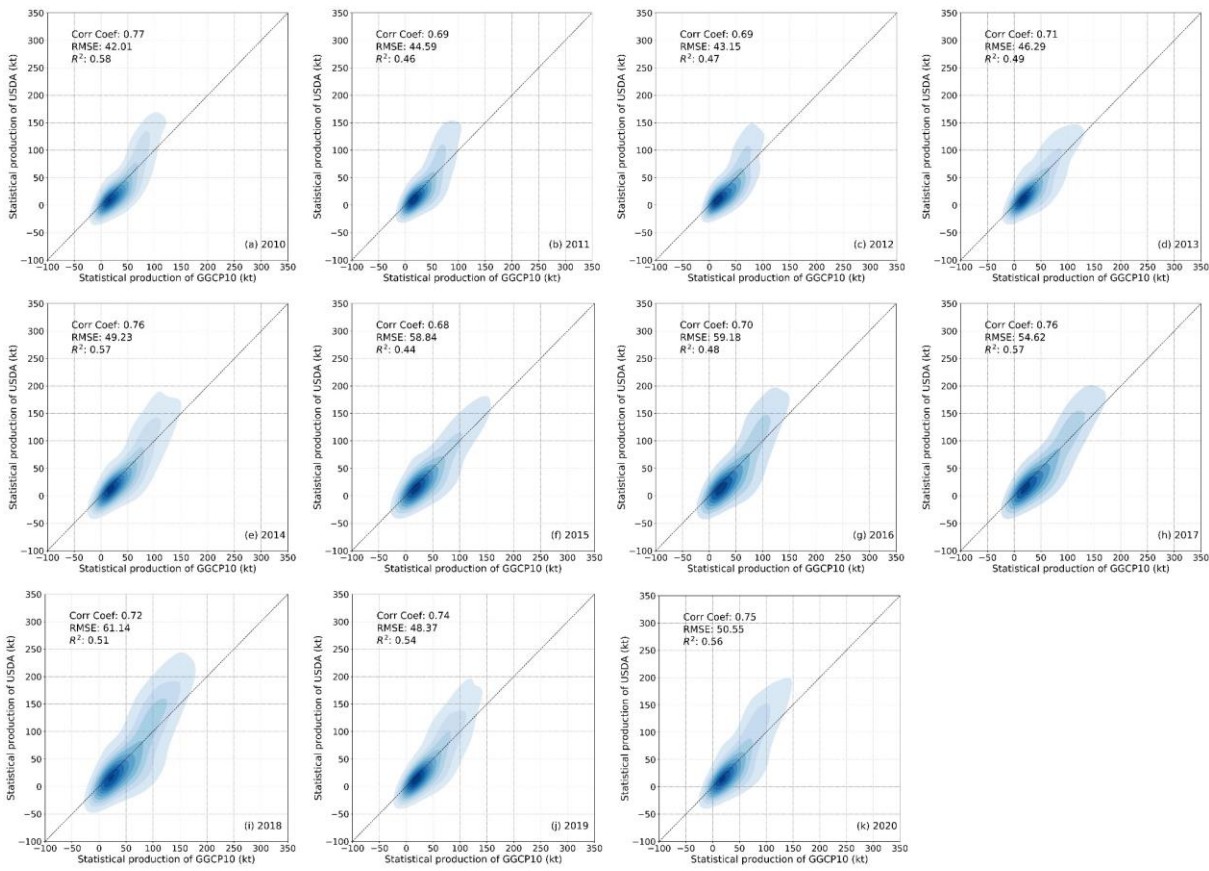

**Figure 15. Kernel density estimation plots of the two datasets of soybean from 2010 to 2020.**


For soybean (Fig. 15), GGCP10 shows a high consistency with the USDA dataset. Although $R^2$ drops to 0.44 in 2015, this is still a relatively high consistency. Further analysis shows that the majority of values in both datasets are concentrated in low-production areas where consistency is very high. For the small number of data points in high-production areas, GGCP10 shows a slight underestimation compared to the USDA data. This may reflect small deviations of our model in handling data for high-production areas, but the overall consistency remains substantial.

The four crops all have relatively high correlation coefficients, showing good agreement between our dataset and the USDA data. Although $R^2$ is relatively lower for wheat, this may be due to potential sampling bias as the USDA dataset is derived from sample surveys. Overall, our dataset shows high consistency with the USDA data, demonstrating its higher reliability and reference value.




After conducting a thorough comparison between the GGCP10 and USDA datasets, we sought to further investigate the discrepancies and potential causes by analyzing the trends in harvested areas from both datasets. Understanding the variations in harvested areas is pivotal in assessing the quality of production data, hence getting insights into these changes can aid in a more comprehensive evaluation

and interpretation of discrepancies in production values. To this end, we present the annual changes in production harvested area, accuracy between GGCP10 and USDA datasets from 2010 to 2020 (Fig. 16).

It's worth mentioning that the USDA harvested area data, along with the production data, is obtained from the "SURVEY" dataset, ensuring consistency in our comparison sources. Regarding our dataset, the harvested area values are intermediate data produced during the development of GGCP10.

To illustrate the differences between the two datasets objectively, we aggregated data from all counties for each year.

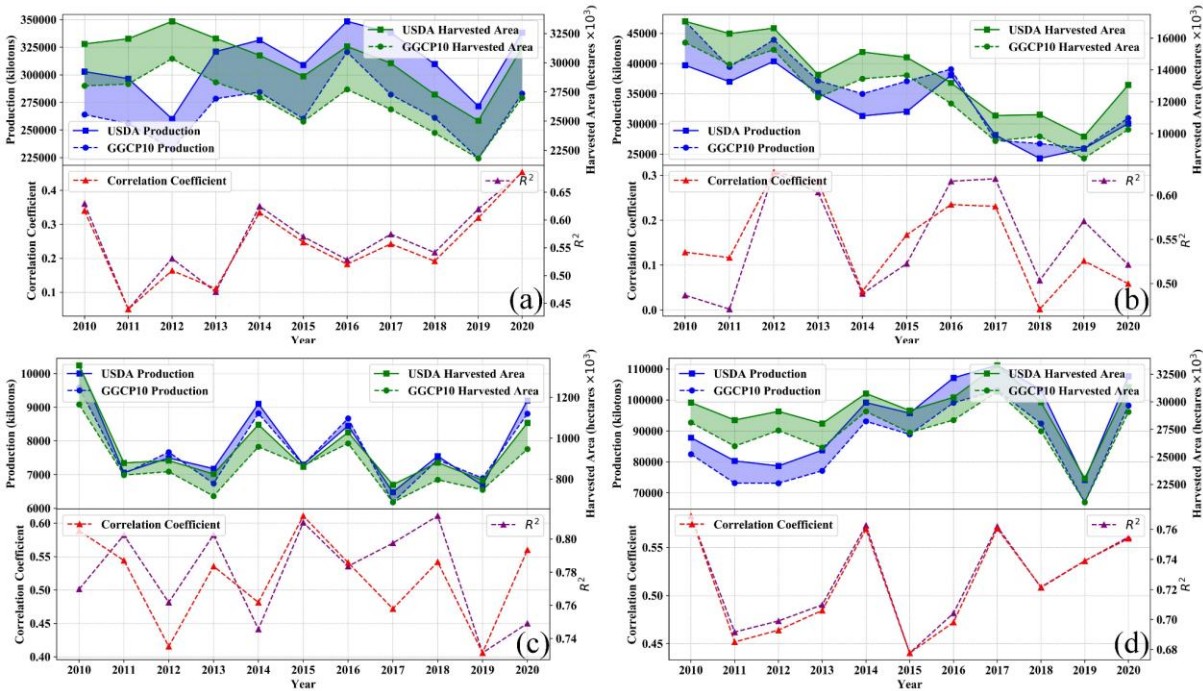

**Figure 16. Annual changes in production, harvested area, accuracy between GGCP10 and USDA datasets from 2010 to 2020: (a) maize; (b) wheat; (c) rice; (d) soybean**



For all four crops, a noticeable pattern emerges: when the discrepancies in annual harvested area between the two datasets widen, the accuracy and production curves tend to diverge more. This is particularly pronounced for maize and wheat, where fluctuations in annual harvested area inconsistency between GGCP10 and the USDA datasets have led to divergent tendencies in both production and accuracy trends. For rice and soybean, the harvested area curves of the two datasets are more closely

aligned, which has resulted in tighter adherence of the accuracy and production change curves.

These findings highlight the significance of maintaining consistent harvested area. As harvested area is a crucial parameter in production estimation, any discrepancies in data between databases would result in deviations in production estimates. Additionally, annual accuracy changes, associated with these area discrepancies, show a direct correlation: the larger the gap in harvested areas, the lower the

accuracy between datasets.

The results in Fig. 16 underline the fact that although GGCP10 is generally consistent with USDA data, the nuances of annual changes in harvested area and the associated inconsistencies are essential factors in understanding and interpreting the reliability of crop production datasets.

**3.5 Advantages and Limitations**

The GGCP10 dataset proposed in this study has considerable advantages over currently available crop production datasets. It is the first continuous (2010-2020) global gridded crop production dataset with temporal resolution of 10 km spatial resolution to our knowledge. The temporal continuity and fine spatial resolution allow for novel investigations of geographic and interannual variations in crop production at regional to global scales(Mohanasundaram et al., 2023).

The time series data allows for analysis of the long-term trends in crop production in response to climate change, shifts in land use, and changes in agricultural policies. The gridded format allows for precise spatial modelling and assessments that were previously restricted by limited and inconsistent agricultural statistics aggregated over large areas(Yu et al., 2020). Examples of research topics in this area include optimization of agricultural resource allocation, evaluation of climate impacts on crop

production, and identification of local production gaps.



However, uncertainties still exist and need addressing in GGCP10. The estimation of harvested area is the primary source of uncertainty. Harvested area data is estimated from area and phenology information. Despite the estimated harvested area's consistency with national statistical data, some level of uncertainty remains at smaller regional scales, particularly for individual grids. The uncertainty stems from our methodology of estimating the harvested area using reference data, cropped area, and phenology information. Numerous factors such as changes in land use/cover, cropping patterns, and climate can impact the estimate's accuracy. This uncertainty will have an impact on the precision of our ultimate estimate of crop production.

In addition, the crop distribution involves a certain degree of uncertainty. Our crop distribution information is based on 2015 reference data. However, the allocation of crop areas may differ in various years as a result of factors such as agricultural policies, market demands, or climate change. This may create uncertainty in crop distribution for certain grids, consequently impacting our estimations of harvested area and crop production(Ramankutty et al., 2008). The comparison with other datasets, as detailed in section 3.4, further highlights some of these limitations. GGCP10 displays considerable consistency with datasets such as that of the USDA, for example, but there are instances of overestimation or underestimation in certain regions or for certain crops.

Therefore, it is crucial to consider the impacts of the aforementioned uncertainties when using the GGCP10 dataset for related research. Firstly, due to the variable nature of harvested area and crop distribution data on an annual basis, these elements could affect the model's sensitivity to climate change, potentially leading to biased research findings. Secondly, as this dataset is adjusted for consistency with FAO's national-level statistics, the source data for different countries primarily come from their respective agencies. This implies that the reliability of the statistical data may vary between countries, and such regional differences in reliability could influence the conclusions drawn from cross-national or large-scale comparative analyses. Moreover, the calculated yield from this dataset may differ from that obtained through ground-based surveys, possibly leading to overestimation or underestimation of yield-affecting factors.

Additionally, this dataset does not consider the instant effects of sudden natural disasters, such as dry hot winds or pest infestations on harvested lands, as these short-term events are typically challenging to capture precisely through large-scale remote sensing. When researching the impact of severe weather events on agricultural production, it should be noted that the data gathered in certain areas may not entirely represent their short-term effects. Furthermore, the lack of crop distribution information gained from field surveys can result in inaccuracies in datasets concerning modifications to cropping patterns driven by policy shifts or alterations in market demand. The precise evaluation of factors affecting agricultural production at either a global or regional scale may be compromised further due to this.

To achieve a more comprehensive and accurate interpretation of results when conducting agricultural research with this dataset, users should take into account the possible impacts of multiple factors mentioned above. We recognise the significance of continually refining and advancing the GGCP10 dataset. By incorporating precise crop distribution data, localizing information, and harnessing developments in remote sensing and machine learning, we aim to enhance the precision and comprehensiveness of future iterations..

## 4 Data availability

The GGCP10 product is available on Harvard Dataverse: https://doi.org/10.7910/DVN/G1HBNK(Qin et al., 2023). It is the first 10km-resolution, temporally continuous, gridded dataset of crop production covering a global extent.

## 5 Conclusion

We propose a global gridded crop production dataset, named GGCP10, which covers the years 2010 to 2020 at a spatial resolution of 10km. This is, to our knowledge, the first temporally continuous, gridded dataset of crop production that is globally available. The GGCP10 involved data from various sources, and a set of data-driven models were developed based on agro-ecological zones and multiple

factors. These models can capture the inherent correlation between crop production, harvested area, and other indicators to achieve high prediction accuracy. The GGCP10 dataset offers a new perspective into the spatial distribution of crop production, which could be valuable for enhancing global food security and promoting sustainable agricultural development, facilitating relevant research, guiding agricultural policies, and enabling multiple applications.


Although our model and dataset exhibit considerable accuracy and reliability, there are still some associated uncertainties, mainly due to the ambiguity of harvested area and crop distribution. When using the dataset, users need to pay close attention to these uncertainties. In the future, we will continue to update and improve this dataset to enhance its usefulness for research in global agricultural and food security.


## Author contribution

XQ, HZ, and BW conceptualized the study and designed the experiments. XQ carried out the experiments and simulations. XQ and HZ developed the model code. BW and HZ were responsible for funding acquisition. BW administered the project. XQ and MZ conducted investigation and formal analysis. MZ and FT performed validation and visualization of results. XQ prepared the original draft of the manuscript. XQ, BW, HZ, MZ, and FT reviewed and edited the manuscript.


## Competing interests

The authors declare that they have no conflict of interest.

## Acknowledgements

During this research, several datasets were crucial. We used information from the FAO Statistical Database and would like to express our gratitude to the Food and Agriculture Organization (FAO) for making it available. Furthermore, we obtained data from the Directorate of Economics and Statistics (DES) of the Deptt. of Agriculture and acknowledge their contributions. Our study also gained from the





data dataset of GLASS, and we are grateful to the GLASS team for their invaluable data. The
GAEZ+2015 dataset was utilized in some of our research, and we thank the GAEZ team and
contributors. Additionally, we integrated crop survey data from the United States Department of
Agriculture (USDA) and are grateful for this essential dataset.

## Financial support

This research was supported by the National Key Research and Development Project of China (No.
2019YFE0126900), Natural Science Foundation of China (No. 41861144019), and ANSO Strategic
Consulting Project (No. ANSO-SBA-2022-02), Agricultural Remote Sensing Innovation Team Project
of AIRCAS (No. E33D0201-6).

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
