# Peer review of "GGCP10: A Global Gridded Crop Production Dataset at 10km Resolution from 2010 to 2020"

_Earth System Science Data, 2023_

## Author Comment (AC1)

Dear Reviewer,

Thank you for your valuable comments and suggestions. Below, we would like to address your concerns point by point.

**Comment 1:**

Spatial distribution of crops is critical information for food security, agriculture development and investment decisions, sustainable agricultural development etc. There have been multiple attempts, by different teams in the world, to produce global crop maps. And yet so far few attempts have been made to produce time series global crop maps. The GGCP10 dataset focuses on maize, wheat, rice, and soybeans and covers the years 2010 to 2020, the first temporally continuous, gridded dataset of crop production at the global scale. The dataset was constructed using a data-driven spatial production allocation model that incorporated multiple source datasets. The use of various data sources, including FAO statistical data, GAEZ+ 2015 annual crop data, and other sources, demonstrates a robust foundation for the study. This model was rigorously examined through pre-processing and consistency checks to ensure data accuracy and reliability. The incorporation of machine learning techniques for predicting crop yields and production is a forward-looking approach. These techniques have demonstrated solid performance in recent years. The approach of combining information from multiple sources, including climate, soil, and topographic data, is a commendable strategy for predicting crop production accurately.

**Response to Comment 1:**

We sincerely appreciate your positive evaluation of our work and your insightful summary of the significance of the paper. Your high appraisal of the importance of this research in constructing a temporally continuous and globally spatially covered gridded crop production dataset, as well as the robustness and innovativeness of the data-driven spatial production allocation model in integrating multi-source datasets, will be a great encouragement for our future research work. We will further refine the paper according to your valuable comments, striving to provide more long-term and accurate data support for global agricultural production mapping and food security research. Thank you again for your detailed review and valuable suggestions. We cordially invite you to continue providing us with your insightful feedback and guidance when we submit the revised manuscript.

**Comment 2:**

My first concern is that their whole modelling approach, production model in particular (see Section 2.2.3 Data-driven Model Training), implicitly assumes that the biophysical parameters alone could determine the crop production. In other words, their modelling approach assumes that the driven factors for the huge spatial heterogeneity of crop productivity (or production if crop area is counted) are mainly those biophysical parameters such as soil, AEZ zones, various vegetation indices, climate variables (multi-source indicators XI(i,j) as shown in their model, Line 225). Any breeders or agricultural economists would tell you that this is not true. Social economic factors such as crop seeds/varieties, crop management, fertilizer, pesticide are the major driven force in crop productivity (and so crop production). This is why, for example, the maize yield in a large estate farm in Zambia could be a few times higher than that of a subsistence maize farmer next door – just a few hundred meters away! Of course collecting the data for these parameters on a global scale is much harder, if possible at all. Without the inputs of these critical parameters, estimating crop yields spatially is a huge challenge.

**Response to Comment 2:**

We appreciate your insightful comments. We agree that socio-economic factors (such as crop management practices) play a crucial role in shaping the spatial heterogeneity of crop production. Obtaining such data on a global scale poses significant challenges, which is one of the limitations of the current research. We will supplement the discussion section of the paper to elaborate on this limitation.

1) It is worth noting that although we lack direct data on crop management practices, **some of the factors included in the model can, to a certain extent, reflect the spatial differences in crop management levels.** For example, irrigation data reflects differences in irrigation management inputs, which greatly influence crop growth and production. Crop planting area data partially reflects farmers' planting preferences and resource allocation decisions for different crops, demonstrating farmers' responses to market conditions and policies.

2) Some of the remote sensing-derived indicators we included can also reflect the impact of crop management to a certain degree. For instance, the Maximum Vegetation Condition Index (VCIx) describes the historical relative level of vegetation conditions during the study period. A higher VCIx indicates relatively better crop growth during that period, which to some extent benefits from farmers' good field management. Indicators such as Net Primary Productivity (NPP) and Leaf Area Index (LAI) reflect crop biomass accumulation and photosynthetic intensity. Higher NPP and LAI are often the result of good management.

3) Moreover, agro-ecological zone data comprehensively considers the impact of natural conditions such as climate, soil, and topography on agricultural production. Different ecological conditions often correspond to differentiated planting systems

and management patterns. Therefore, one important reason we chose to model at the agro-ecological zone scale is that we hope the model can characterize the spatial variation of production within the ecological zone through the differences in multiple variables within the zone.

4) Of course, we also recognize that due to data limitations, the current model does not incorporate key crop management factors such as cropping systems, variety selection, and fertilizer use, which may obscure some important drivers of production variation. Therefore, in the discussion section, we will further analyze this limitation of the model and its impact on data quality.

**Comment 3:**

My second concern is that the paper is a data description paper and yet it misses the critical dataset: a global sub-national crop statistics data. Their major statistical data source is the FAOSTATA data at country level, which is too coarse for the gridded product. Crop type mapping is too complex and too dynamic to be able to be modeled without the actual sub-national statistics. For example, farmers may decide to reduce their maize area and instead plant more rice in the current season if they expect more rain in the coming season or simply they believe the maize price will go down next year. Any fancy modelling approach is difficult to capture that without the actual data. The paper itself emphasizes a lot on their modelling approach while ignoring the time-consuming effort of collecting crop data for the four crops (maize, rice, wheat and soybean). I would say the latter is much more critical, in particular considering that the ESSD journal is, which I quote, "for the publication of articles on original research data (sets), furthering the reuse of high-quality data of benefit to Earth system sciences".

**Response to Comment 3:**

1) We strongly agree with your view on the importance of crop statistics at the national/regional level. However, we would like to further clarify that in this study, **national-level statistics are mainly used for post-processing of model results**, i.e., calibrating the gridded production estimates with national official statistics through consistency processing to ensure statistical consistency of the estimates at the administrative unit level. During the model training and prediction stages, we mainly use spatialized multi-source data such as remote sensing-derived indicators, meteorological and soil data, etc., which can provide high-resolution crop growth information to support fine-scale production mapping. Therefore, the spatial resolution of statistical data does not directly affect the modeling accuracy.

2) Indeed, obtaining crop statistics at sub-national administrative levels is of great importance for understanding farmers' planting behaviors and assessing the impact of regional agricultural policies. To this end, we have made every effort to collect state/provincial-level crop statistics from major agricultural countries such as the United States, Canada, Argentina, Brazil, India, China, Thailand, and Australia

through official channels and partnerships. However, as the reviewer mentioned, these data vary greatly in terms of access channels, update frequency, and spatiotemporal coverage, making it difficult to fully unify the format and connotation of data from different countries. If they are used for consistency calibration, it may be difficult to ensure statistical consistency at the national scale.

3) In contrast, **the national agricultural statistics released by the FAO have obvious advantages**. Although the spatial resolution is coarser, the data sources are authoritative, the time series is long, and the official statistics of various countries have been systematically summarized and verified. At the current stage, using FAO data for global-scale production estimation result calibration can maximize the use of existing data resources while ensuring the consistency of data benchmarks across countries. Such global production mapping data based on a unified benchmark can better serve applications such as monitoring the SDGs and assessing global food security. Of course, in the long run, agricultural statistics at sub-national administrative levels are irreplaceable for understanding regional agricultural production processes and optimizing resource allocation.

4) In addition, we would like to further emphasize that the collection, production and processing of the large amount of basic data used in this study is very time-consuming. In addition to using publicly published data products, the potential biomass, CALF, and VCIx indicators we use require the research team to go through a series of complex processing steps from raw data acquisition to final indicator generation, including data storage, format conversion, radiometric calibration, atmospheric correction, geometric correction, cloud and snow masking, vegetation index calculation, and pixel compositing. Each step requires professional algorithm design and parameter tuning to ensure data quality.

**Comment 4:**

The paper could benefit from more transparency regarding data preprocessing steps, such as how data clipping based on crop phenology is conducted and how missing or corrupted data are handled. For example, Line 179-183, Where does CA(i,t) come from? How to divide CA(i, t) into CA(i,j) ? Not clear at al. I think (I am not 100% sure as I have a hard time to understand this section) "reference year" at Line 184 should be "target year". After reading the section multiple times, I still don't know how the harvested area is estimated at the pixel level. I considered myself as an expert, imagine how an ordinary reader would feel!

**Response to Comment 4:**

Thank you for your thorough review and constructive suggestions. We will follow your advice to provide a clearer and more detailed description of the data preprocessing steps in the data and methods section, so that readers can accurately understand each processing step.

1) First, **regarding data clipping based on crop phenology, we adopt a method of extracting time windows and calculating feature indicators from time-series data based on crop phenology information**. Specifically, we first determine the time range of the main growth stages (such as sowing, growing and maturity) for each crop type according to the crop phenology; then, we extract the corresponding time-series data based on the main growth stages of each crop as time windows; finally, within each time window, we calculate the statistical feature values (such as maximum, minimum, standard deviation and total sum) of the time-series indicators (such as cumulative precipitation, average air temperature) and use them as input features for the model. For the processing of missing or abnormal data, we will also provide detailed technical details in the revised manuscript to facilitate readers' understanding and improve the transparency and reproducibility of the data processing methods.

2) Secondly, **regarding the lack of clarity in the description of harvested area estimation** at the pixel level in lines 179-183 and the surrounding context. The estimation of harvested area is one of the important innovations in this study. We combine the gridded data (harvested area, planted area) of the reference year, the gridded data (planted area) and statistical data (harvested area) of the target year to dynamically estimate the harvested area of each grid within the region. Due to the limited space, we will introduce the principles and processes of harvested area estimation in detail in the revised manuscript, striving to clearly describe this innovative method so that both experts and ordinary readers can better understand it.

3) **Regarding other detailed issues you mentioned**, such as $CA(i,t)$, $CA(i,t)$ represents the total planted area of all crops in the i-th grid in the t-th growing season, which is obtained by multiplying the cropped arable land fraction (CALF) of the i-th grid by the cropland area of that grid. For line 184, "reference year" should indeed be "target year". We will carefully proofread and refine this part of the content in the revised manuscript.

Thank you again for your valuable comments. We will revise the paper accordingly to meet the publication requirements.

**Comment 5:**

Model Selection: The paper mentions the selection of machine learning models but lacks specific details about the criteria used for model selection. Providing more insight into the model selection process would enhance the paper's transparency.

**Response to Comment 5:**

We greatly appreciate your suggestion. We agree that providing more details in the model selection section will help improve the transparency of the research. We will expand and refine this part of the content. Specific revisions include:

1) First, in the revised manuscript, **we will explain in detail the reasons for choosing machine learning models**. Compared to traditional statistical models, machine learning models have advantages in dealing with complex nonlinear relationships and high-dimensional data. At the same time, our previous research has shown that machine learning models perform well in crop production estimation[1]. Therefore, we adopt machine learning models to construct the spatial allocation relationship of production. Regarding the selection of Random Forest, XGBoost, and CatBoost as candidate models, it is mainly based on their good performance and robustness in the field of geospatial modeling: Random Forest is one of the most commonly used algorithms in ensemble learning; XGBoost has achieved excellent results in various data mining competitions and has been proven to have significant advantages in processing high-dimensional and nonlinear relationship data; CatBoost has shown outstanding performance in multiple data science competitions and is considered a powerful tool for handling mixed data. We will provide a more detailed explanation of the reasons for model selection in the revised manuscript.

   [1] Li, Y., Zeng, H., Zhang, M., Wu, B., Zhao, Y., Yao, X., Cheng, T., Qin, X., and Wu, F.: A county-level soybean yield prediction framework coupled with XGBoost and multidimensional feature engineering, International Journal of Applied Earth Observation and Geoinformation, 118, 103269, https://doi.org/10.1016/j.jag.2023.103269, 2023.

2) Secondly, **regarding the specific process of model selection**, we adopt a nested cross-validation strategy to optimize the hyperparameters and evaluate the performance of the three models. Specifically, we first divide the samples into a training set and a test set. On the training set, we use 5-fold cross-validation to perform grid search optimization on the model hyperparameters. By traversing different hyperparameter combinations, such as the number of trees and maximum depth of Random Forest, learning rate and number of trees of XGBoost and CatBoost, etc., we find the optimal hyperparameter configuration for each model. Then, we retrain the models using the optimized hyperparameters and evaluate the prediction performance of the models on the test set. The model performance evaluation metrics is coefficient of determination ($R^2$), which measure the model's ability to explain production variations. We select the model with the highest $R^2$ on the test set as the optimal model for that agro-ecological zone. It should be emphasized that the above hyperparameter optimization and model evaluation process is carried out independently within each agro-ecological zone to obtain the optimal model for the characteristics of different regions. This strategy helps to improve the model's ability to characterize regional production variation

characteristics and enhance the accuracy of regional production estimation. In the revised manuscript, we will also display the optimal model for each region in a spatialized manner (as shown in **Figure 1**) to intuitively show the advantageous distribution of different models in each agro-ecological zone, aiming to reveal the association between model selection and regional characteristics, and explore the potential relationship between model applicability and regional factors such as climate, soil, topography, and planting systems. It should be noted that this figure is intended to intuitively show the spatial distribution pattern of model selection results. To simplify the illustration, we have not shown national boundaries. For countries without subdivided agro-ecological zones, model selection is performed at the national scale.

[Figure]

**Figure 1. Spatial distribution of optimal model selection results.**

3) In addition, if the paper is finally published, we commit to providing complete model training and evaluation code, as well as detailed code documentation, which will help readers fully understand our modeling process and reproduce or improve our methods in other studies.

Through the above supplements and improvements, we believe that the transparency of the model selection section will be greatly enhanced, allowing readers to better understand and evaluate our modeling ideas. Thank you again for your valuable suggestions.

**Comment 6:**

Data Limitations: While the paper discusses data limitations briefly, a more thorough exploration of potential data limitations, such as inaccuracies in remote sensing data or potential biases, would provide a more comprehensive view.

**Response to Comment 6:**

We strongly agree with the your viewpoint that a comprehensive discussion of data limitations is valuable for objectively evaluating research results and guiding future research. We will expand and deepen this part in the revised manuscript.

1) **First, regarding the uncertainty of remote sensing data**, we will focus on the following points: Firstly, the quality of remote sensing data is affected by factors such as sensor performance, atmospheric conditions, and surface heterogeneity. In cloudy and rainy areas, clouds and fog will obscure the surface, resulting in missing or decreased quality of data at some spatiotemporal resolutions. In cold high-latitude regions, winter snow cover also affects the extraction accuracy of surface parameters. Secondly, errors may be introduced in the data processing steps such as radiometric calibration, atmospheric correction, and geometric registration, affecting the accuracy of the data. Furthermore, the mixed pixel problem may lead to insufficient representativeness of the extracted surface parameters, especially in regions with severe fragmentation of farmland. These factors may cause systematic biases in remote sensing products in individual regions or time periods, such as overestimating vegetation indices during the growing season.

2) **Secondly, regarding the potential biases in statistical data**, we will focus on the following aspects: First, differences in statistical caliber and standards may lead to insufficient comparability of data between different countries and regions. Second, statistical data may have biases such as underreporting, misreporting, and missing reports, which affect the reliability of the data.

3) Third, **we will provide detailed comparisons between GGCP10 and reference datasets to further explore the strengths and weaknesses of the GGCP10 data** by conducting spatial consistency analyses. For example, in **Figure 2** below, GGCP10 exhibits better spatial transitions compared to SPAM 2010; in **Figure 3**, GGCP10 and USDA survey data show high consistency in high-production counties, while in low- production counties, overestimation or underestimation may exist. In the revised manuscript, we will conduct more detailed analyses to better understand these patterns.

[Figure]

**Figure 2. Spatial comparison of crop production between SPAM 2010 and GGCP10 datasets for selected regions: (a) Maize production in Africa; (b) Wheat production in Western Europe; (c) Rice production in Southeast Asia; (d) Soybean production in Brazil and Argentina, South America.**

[Figure]

**Figure 3. Spatial comparison of Soybean production between USDA survey data and GGCP10 at the county level for the years 2010, 2015, and 2020.**

4) **Then, we will also explore the scale effect and spatiotemporal matching issues**. Production statistics are mostly at the administrative unit scale, while model input data are at the pixel scale, and the scale difference between the two may cause errors. The spatiotemporal resolution and boundary definitions of different data products may also not be completely consistent, affecting the accuracy of data matching and fusion. To reduce errors caused by inconsistent administrative boundaries, we consistently used the standard administrative boundary data provided by the FAO in data processing. On the other hand, the spatial resolution of our production data is 10 kilometers, which has a certain scale difference compared to the county-level scale of statistical data. This may lead to a certain averaging effect of the estimated county-level production, resulting in overestimation or underestimation when compared with county-level statistical production data.

5) Finally, we fully understand that **users may encounter situations where our production dataset is not completely consistent with local statistical data** when using it. To address this issue, we will suggest the following processing strategies for users in the revised manuscript: First, it should be recognized that our data may have systematic overestimation or underestimation in individual regions, but overall, it can better reflect the spatial distribution differences of production within the region. Secondly, if users have more reliable regional statistical data, we recommend that users use these data for secondary calibration of our initial estimation results. Specifically, users can calculate the ratio coefficient between our initial estimates and the regional statistical totals, and then use this coefficient to proportionally scale the gridded production data to match the regional statistical totals. This post-processing method not only preserves the spatial distribution information of production revealed by our data but also utilizes local data to improve the accuracy of regional total estimation.

Through the above discussion, we will fully respond to your concerns about data limitations, making the paper's discussion on data quality and applicability more comprehensive and objective. Thank you again for your valuable comments. We sincerely invite you to review the revised manuscript once we submit it and kindly request your continued feedback and suggestions.

---

## Author Comment (AC2)

Dear Reviewer,

Thank you for your valuable comments and suggestions. Below, we are providing a point-to-point response to each comment.

**Comment 1:**

Qin et al. used multiple data sources, including statistical data, gridded production data, agroclimatic indicator data, agronomic indicator data, global land surface satellite products and ground data, to develop a data-driven crop production spatial allocation model, and generated the global 10km resolution gridded production dataset of four major crops (maize, wheat, rice and soybean) from 2010 to 2020. Basically, this topic is necessary. However, this study has several serious issues.

**Response to Comment 1:**

Thank you for recognizing the importance of this research work and affirming the innovations in data and models. At the same time, we greatly value the issues and suggestions you have raised. These comments provide important insights for us to further refine and improve the research ideas, methods, and quality of the results. We will carefully analyze and incorporate your valuable feedback and make every effort to address them in the revised manuscript. We look forward to receiving your review comments and guidance again after the completion of the revised manuscript to further enhance the academic and application value of the paper. Thank you again for your pertinent suggestions.

**Comment 2:**

First, the method for generating production map is not robust. The authors used an existing production map as the reference and training a machine learning model to allocate statistical production to grids. The method is not innovative, and is unreasonable from the importance analysis of input features (see my below comment). In principle, there is no reason to prove the machine learning method work here, because the planting area of a given pixel can not be predicted at all which depends on the famers' activities. Therefore, I did not believe this method can work for generating production map globally or regionally.

**Response to Comment 2:**

1) We greatly appreciate your insightful comments. You have raised a critical

challenge in production mapping research, which is how to effectively establish a quantitative relationship between production and environmental factors. Traditional parameterization methods often struggle to fully capture the non-linearity and multi-scale effects of the production formation process. In contrast, the machine learning method employed in this study has unique advantages in this regard. **By constructing complex non-linear models, machine learning methods can better capture the response patterns of production to changes in environmental factors and explore the key influencing factors in the production formation process.** This data-driven modeling paradigm has been successfully applied in the field of agricultural remote sensing, such as [1-5]. We will add corresponding references in the revised manuscript to demonstrate the effectiveness and successful application of such modeling methods. Furthermore, the large amount of multi-source input data used in this study, including key indicators such as maximum vegetation condition index, and cropland arable land    fraction, are unique features developed by the research team. Moreover, to better adapt to the differences in agricultural planting systems in different regions, we have adopted an agro-ecological zoning modeling strategy, where model training and parameter optimization are performed independently within each zone. This modeling concept helps to improve the model's ability to characterize regional features. Therefore, the method used in this study is reasonable and innovative.

[1] Feng, P., Wang, B., Liu, D. L., Waters, C., and Yu, Q.: Incorporating machine learning with biophysical model can improve the evaluation of climate extremes impacts on wheat yield in south-eastern Australia, Agricultural and Forest Meteorology, 275, 100–113, https://doi.org/10.1016/j.agrformet.2019.05.018, 2019.

[2] Franz, T. E., Pokal, S., Gibson, J. P., Zhou, Y., Gholizadeh, H., Tenorio, F. A., Rudnick, D., Heeren, D., McCabe, M., Ziliani, M., Jin, Z., Guan, K., Pan, M., Gates, J., and Wardlow, B.: The role of topography, soil, and remotely sensed vegetation condition towards predicting crop yield, Field Crops Research, 252, 107788, https://doi.org/10.1016/j.fcr.2020.107788, 2020.

[3] Kamir, E., Waldner, F., and Hochman, Z.: Estimating wheat yields in Australia using climate records, satellite image time series and machine learning methods, ISPRS Journal of Photogrammetry and Remote Sensing, 160, 124–135, https://doi.org/10.1016/j.isprsjprs.2019.11.008, 2020.

[4] Ma, Y., Zhang, Z., Kang, Y., and Özdoğan, M.: Corn yield prediction and uncertainty analysis based on remotely sensed variables using a Bayesian neural network approach, Remote Sensing of Environment, 259, 112408, https://doi.org/10.1016/j.rse.2021.112408, 2021.

[5] Li, Y., Zeng, H., Zhang, M., Wu, B., Zhao, Y., Yao, X., Cheng, T., Qin, X., and Wu, F.: A county-level soybean yield prediction framework coupled with XGBoost and multidimensional feature engineering, International Journal of Applied Earth Observation and Geoinformation, 118, 103269, https://doi.org/10.1016/j.jag.2023.103269, 2023.

2) Regarding the issue of predicting farmers' planting areas, we believe that our unclear description may have caused some misunderstanding. When modeling, we do not directly use natural factors to predict farmers' planting areas. Instead, **based on the remote sensing observed planting area data of the reference year, we dynamically update it using interannual change information, which**

**is an important innovation of our method.** We used the title "harvested area estimation" but the actual method adopted is a dynamic area calculation method, i.e., using the change in cropland utilization intensity (the proportion of cropland planted within the grid) observed by remote sensing to characterize the spatiotemporal dynamics of planting area. This strategy, to a certain extent, reflects farmers' dynamic responses to market and policy signals. However, our description of this method in the manuscript may not be detailed and comprehensive enough, which may have led to some misunderstandings. In the revised manuscript, we will further clarify the principles and innovations of the planting area estimation method.

3) We agree that due to the lack of socio-economic behavior data at the farmer scale, the current model cannot directly characterize the impact of farmer decision-making on planting structure, which is a limitation of this study. We will explain this in the discussion section. However, we believe that by integrating environmental factor data and knowledge of crop growth processes, the existing model can still well reveal the spatiotemporal patterns of regional production, which is of great value for understanding the geographical differentiation of food production.

**Comment 3:**

Second, the harvested area map used by the authors too simple to indicate the spatial and temporal variations. As we know, the most important feature for production is harvested area. However, the authors assumed a fixed ratio to exact the harvested area from a given year to other years. I did not think the harvested area map can reproduce the spatial and temporal changes, which is still static like the previous study. And anyone can easily generate production map based on this assumption without as an input of harvested area.

**Response to Comment 3:**

1) We sincerely appreciate your valuable comments. Characterizing the spatiotemporal dynamics of harvested area is critical in production mapping. Based on this understanding, we did not adopt a simple fixed ratio method in this study. Instead, we fully utilized the cropland utilization dynamics information provided by remote sensing observations to dynamically update the harvested area on an annual basis. **This method not only considers the temporal changes in cropland planting intensity but also characterizes the spatial heterogeneity of these changes**. It is an important improvement and complement to the traditional static area estimation method. This is also a key innovation of this study.

2) In the harvested area estimation process, we fully considered the spatial heterogeneity and dependence of harvested area changes. At the agro-ecological

zone scale, we estimated the harvested area using spatial statistics and geographically weighted methods. The related methods have been described in detail in Section 2.2.1 of the manuscript. We realize that in the preprint, the description of the above methods may not be detailed and comprehensive enough, which may affect readers' understanding of the innovation points. In the revised manuscript, we will supplement the detailed description of the estimation principles and key technical routes, accompanied by flow charts and other visual presentations to improve the transparency and reproducibility of the methods.

**Comment 4:**

Third, the writing still need a lot of work. I am always confused that the authors exactly mean. And there are a lot of places that the authors missed necessary details which made the manuscript hard to follow.

**Response to Comment 4:**

1) We sincerely appreciate your valuable comments. We will carefully revise and proofread the entire manuscript according to your suggestions, striving to eliminate ambiguities in expressions and make the article more concise, coherent, and readable.

2) Regarding the lack of necessary details mentioned by the reviewer, we will focus on improving the following aspects: First, we will supplement key information such as experimental schemes, technical routes, and model assumptions in different sections of the article according to their focus, providing readers with necessary background knowledge. Second, in the data and methods section, we will provide a more detailed description of the data processing workflow, model parameter selection, and evaluation metric definitions to make the research process more transparent and standardized.

3) Third, we will enhance the correspondence between figures, tables, and text descriptions by adding table headers, legends, and variable annotations to improve the self-explanatory nature of the figures and tables, making it easier for readers to understand. Fourth, when professional terms and algorithm names appear for the first time in the text, we will provide clear definitions or explanations to avoid ambiguity. We will also invite native English-speaking peers to polish the article.

**Comment 5:**

2.1.9 section: the authors should introduce the indicators first. I do not understand what cumulative potential biomass means, which is a satellite-based observation or a predicted variable. The authors provide a reference which a website in Chinese and

the readers can not find accurate definition from there. Same problem also was found in the section 2.1.10, like VCIx. By the way, there are a lot of places with this kind unclear writing issues making the reading very hard.

**Response to Comment 5:**

1) We sincerely apologize for the lack of clarity and necessary explanations of the indicators mentioned in the manuscript. We will prioritize improvements in the revised manuscript. Specifically, in Section 2.1 "Data," when introducing each indicator, we will first provide a clear definition, explaining its basic concept, mathematical expression, and ecological significance, providing readers with necessary background knowledge and corresponding references.

2) Taking the cumulative potential biomass (BIOMASS) and maximum vegetation condition index (VCIx) you mentioned as an example, we will supplement the following explanation in the main text:

"Cumulative potential biomass is expressed as the combined effect of rainfall (Rain) and temperature (Temp) accumulated during a reference period (dekad from i to n) using the following equations:

$$NPP_{Rain} = \sum_{dek=i}^{n} NPP[Rain(dek)]/n$$

$$NPP_{Temp} = \sum_{dek=i}^{n} NPP[Temp(dek)]/n$$

$$NPP = \min (NPP_{Rain}, NPP_{Temp})$$

The unit of biomass is grams of dry matter per square meter over the concerned period.

Based on the Vegetation Condition Index (VCI) proposed by Kogan (1990), the maximum VCI is adopted in CropWatch bulletins to describe the optimal crop condition of the current period compared with the historical maximum crop biomass potential using the following equation:

$$Maximum\ VCI = \frac{NDVI_{max\_c} - NDVI_{min\_h}}{NDVI_{max\_h} - NDVI_{min\_h}}$$

where $NDVI_{max\_c}$ is the maximum NDVI of a fixed period, and $NDVI_{max\_h}$ and $NDVI_{min\_h}$ are the historical maximum and minimum NDVI of the same period, respectively, using long-term time series NDVI datasets. Considering that the crop minimum NDVI may be contaminated by clouds or non-vegetation pixels , an empirical minimum vegetation NDVI value (0.15) is introduced to calculate $NDVI_{min\_h}$ with the following equation:

$$NDVI_{\min\_h} = \max(0.15, NDVI_{\min\_h0})$$

where $NDVI_{\min\_h0}$ is the original minimum NDVI of the study period from time series NDVI datasets. The value of Maximum VCI ranges from 0 to 1. A higher maximum VCI value indicates better crop condition and larger biomass potential for a concerned period. Therefore, crop maximum VCI is more meaningful when calculated during the crop growing period. "

3) For other indicators mentioned in the text, we will provide detailed explanations following the above approach.

4) To facilitate readers' better understanding of the indicators involved in the study, we will systematically review the references and websites and provide authoritative English definitions and algorithm descriptions whenever possible. For the few indicators that lack English literature support, we will provide more detailed concept explanations and calculation steps in the main text to ensure clarity and completeness of the descriptions. At the same time, we will carefully check the entire manuscript and strive to be professional, rigorous, and standardized in our writing expressions to eliminate language barriers that may affect readers' understanding.

**Comment 6:**

Line 167: it is very difficult to understand what you mean here.

**Response to Comment 6:**

We sincerely appreciate your pointing out this expression issue and apologize for the reading barriers caused by it. We will improve it in the revised manuscript.

1) Specifically, line 167 originally read: "However, this approach ignores changes in cropping conditions between different grid cells within a region." Our intention was to say that the simple proportional allocation method assumes that the changes in planting structure of each grid within a region are consistent, ignoring the spatial heterogeneity of crop planting conditions between grids. Specifically, due to the spatial variability of natural conditions and agricultural management, there are often significant differences in planting structure within a region. Directly using the change ratio at the regional level makes it difficult to accurately characterize the dynamics of planting area at the grid scale. Therefore, in response to the above problem, this study proposes a dynamic method that combines the changes in cropland planting area to obtain a more accurate grid-level harvested area. **Compared with traditional methods, this method fully considers the spatial heterogeneity within the region and can more accurately map the spatiotemporal dynamics of planting patterns.**

Thank you again for your valuable comments. In the revised manuscript, we will further emphasize the advantages and innovations of our method and restate the above content in a more concise and accurate language to improve the readability of the article.

**Comment 7:**

Line 169-171: again, I am confused and did not understand the meaning. You may show the correlation between GAZE and CropWatch.

**Response to Comment 7:**

1) We sincerely appreciate your raising this question. Our description of the relationship between GAEZ+ and CropWatch data in the manuscript may not be clear enough, causing confusion in your understanding. Regarding the expression in lines 169-171, we will reorganize the writing context of this part in the revised manuscript, adding necessary background and data explanations to ensure that each detail is accurately and clearly stated.

2) Specifically, the core information we want to express in lines 169-171 is that through a multi-scale correlation analysis of GAEZ+ 2015 harvested area data and CropWatch cropped arable land fraction (CALF) data, we found that the gridded harvested area has a significant positive correlation with the contemporaneous CALF. This finding provides a basis for us to use CropWatch's CALF data to dynamically update the harvested area data.

3) To make this finding more intuitive and understandable, we will provide detailed correlation analysis results in the supplementary materials. We calculated the Pearson correlation coefficients between the two datasets at two spatial scales: country and agro-ecological zone. At the country scale, the average correlation coefficients for maize, wheat, rice, and soybean are 0.49, 0.51, 0.44, and 0.48, respectively, with spatial distributions shown in **Figure 1**. Only a few countries show negative correlations. At the agro-ecological zone scale, the average correlation coefficients for the four crops are 0.47, 0.47, 0.40, and 0.42, respectively, with spatial distributions shown in **Figure 2**. Only a small number of regions show negative correlations. Overall, the correlation coefficients between gridded harvested area and CALF are relatively high, especially in major food-producing areas. These results statistically verify the high consistency between the two datasets, indicating the feasibility of using CropWatch's CALF data to estimate harvested area data.

[Figure]

**Figure 1. Spatial distribution of correlations between GAEZ+ 2015 harvested area data and CALF data at the national scale.**

[Figure]

**Figure 2. Spatial distribution of correlations between GAEZ+ 2015 harvested area data and CALF data at the AEZ scale.**

4) Thank you again for your valuable suggestions. By supplementing the comparative analysis of GAEZ+ and CropWatch data and visualizing the correlation research results, we believe we can more fully demonstrate the theoretical basis, technical route, and innovative features of the harvested area estimation method in this study.

**Comment 8:**

Section 2.2.1: the method for estimating harvested area is not robust totally. The authors assumed the same change rates of all crops with the total cultivation area and used the statistical area by FAO (national level) to estimate harvested area of each grid. It is obviously wrong method, and which may induce large bias among different

years. If the authors did this kind estimation for harvested area why did not just estimate production directly. Besides, the writing of method section need improve and it will be good to introduce the principle first and then write out the algorithm, which will be easier for the readers.

**Response to Comment 8:**

We sincerely appreciate your constructive suggestions. Due to our lack of accurate and comprehensive descriptions in the manuscript, some misunderstandings may have arisen, for which we deeply apologize. **In fact, when estimating the harvested area, we did not simply assume that all crops have the same change rate as the total cropland area. Instead, we performed separate processing for each crop type**.

1) Specifically, we first obtained the total harvested area of major crops in each country from FAO statistical data; then, using the crop spatial distribution weights provided by GAEZ+ data, we decomposed the national totals to each grid cell; on this basis, we further utilized the interannual change information of cropland area reflected by CropWatch data to dynamically correct the initial gridded harvested area. It is worth mentioning that we also fully considered the differences in climatic conditions and planting systems of different countries, incorporating the crop phenology information of each country into the estimation process to further improve the refinement level of the estimation from the temporal dimension. Due to the limited length of the article, our description of these details in the preprint may not be sufficient, which may have led to some misunderstandings. In the revised manuscript, we will focus on strengthening this part of the content to eliminate possible biases in understanding.

2) Regarding the question of why we estimate harvested area separately, it is mainly based on the following considerations: On the one hand, harvested area data itself contains rich agricultural production information, such as multiple cropping index and rotation patterns, which are key factors in assessing regional agricultural planting structure and land use intensity, and have important research value; on the other hand, production is the product of harvested area and yield, and estimating harvested area separately helps us more clearly understand the relative contributions of these two factors to production changes and deepen our understanding of the production composition mechanism. Therefore, although directly estimating production is technically feasible, we believe that separately estimating harvested area is still a necessary and valuable fundamental work. In fact, the harvested area data estimated in this study is not only used for subsequent production mapping but can also support research in other related fields, such as land use change analysis and agricultural policy assessment.

3) Regarding the writing issue of the methods section, we fully agree with you suggestion. We will adjust the current writing approach and, when introducing the research methods, first explain the theoretical basis and main assumptions of the method, and then systematically explain the technical implementation steps in the

form of an algorithm flow chart. In this process, we will also pay attention to supplementing necessary formulas and parameter explanations to enhance the rigor and reproducibility of the method description.

**Comment 9:**

Line 214: a typical writing error 'First, the time series data are clipped by crop phenology to obtain the data corresponding to the crop growth period.' How can the series datasets are clipped by crop phenology? Is it separated into different seasons? Or separate various crop types according to their own phenology? However, these series datasets also include location, terrain or soil according to table 1. How to clip, and why clip these features?

**Response to Comment 9:**

We sincerely appreciate your comments. As you mentioned, the expression "clipping" is indeed not rigorous and accurate enough, which can easily cause misunderstandings. We will follow your advice to carefully revise this part of the content in the revised manuscript, striving to accurately and clearly describe the processing of time-series data.

1) In fact, **the method we adopted is not "clipping" but extracting time windows and calculating feature indicators from time-series data based on crop phenology information.** Specifically, we first determine the time range of the main growth stages (such as sowing, growing and maturity) for each crop type according to the crop phenology; then, we extract the corresponding remote sensing, meteorological, and other time-series data based on the main growth stages of each crop as time windows; finally, within each time window, we calculate the statistical feature values (such as maximum, minimum, standard deviation and total sum) of the time-series indicators and use them as input features for the model. The reason for performing this data processing is based on the following considerations: First, crops have different sensitivities and response mechanisms to environmental conditions at different growth stages. Selecting environmental factor data from specific growth stages helps to improve the correlation between environmental factors and production; second, there are differences in phenology between different crop types. Separately extracting environmental factor data for key growth stages of each crop type can highlight the differences in crop responses to environmental conditions. Therefore, through the definition of crop phenology, we have achieved the matching of environmental factor time-series data with key growth stages of crops and constructed a time-window-based indicator system for different crop types.

2) As for the location, terrain, soil, and other time-invariant environmental factor data in Table 1, our processing method is to directly use their original values or standardized values as input features for the model, without time-window

processing. The reason for listing them in the summary table of time-series data in Table 1 is that they, together with the time-series data, constitute the input feature set of the production estimation model. This point may not be clearly expressed in the text, leading to some ambiguity.

3) Thank you again for your valuable comments. We will follow the above ideas to revise the relevant expressions in line 214 and its context to more accurately and systematically describe the processing of time-series data.

**Comment 10:**

Table 1: the title of table is just 'input features'? what does 'dimensions' mean? What does the total dimension indicate?

**Response to Comment 10:**

We sincerely appreciate your careful review and pertinent comments on the table content. We will carefully revise Table 1 to improve the quality of information transmission in the table. The main purpose of this table is to list the types of input data used for model training, their temporal resolution, and the extracted features and their dimensions. Through the table, readers can clearly understand the basic attributes of each type of input data and the dimensions of the feature vectors extracted from the raw data.

1) First, we will modify the table title to more accurately and comprehensively summarize the main content of the table. **The new table title is proposed as "Table 1. Input data and extracted features used for model training."** This title clearly states that the variables listed in the table are the input data and features of the crop production estimation model, providing necessary background information for readers to understand the table content. At the same time, this title also echoes the relevant descriptions of input data and feature engineering in the main text, which helps to enhance the logical consistency of the entire article.

2) Second, we will optimize and adjust the column names of the table to express the meaning of each column more accurately and concisely. For example, change the column name of the second column from "Feature type" to "Data type" to clarify that this column represents the type of input data (such as annual data, time-series data), rather than the type of extracted features. Change the column name of the third column from "Images per year" to "Temporal resolution" to highlight that this column reflects the temporal resolution information. Below the table, we will also add brief table notes to provide necessary explanations of the main content represented by the rows and columns of the table, helping readers better understand the organizational logic and information structure of the table.

3) Regarding the issue of "dimensions," we will add footnotes in the table to strictly

define the connotation and measurement units of this term. Specifically, "dimensions" refer to the length of the feature vector extracted from each type of input data, which determines the representation capability of that type of data in the model's feature space. Finally, regarding the total dimension, it refers to the total number of dimensions of the feature space obtained by summing the feature dimensions of all input data, reflecting the breadth and depth of information covered by the model inputs. In this study, we used 13 categories and 41 dimensions of input features, which comprehensively characterize the biophysical mechanisms of the crop production formation process. In the revised manuscript, we will further consider and refine the connotation, measurement, and representation of the total dimension information to provide readers with more standardized and clear model complexity information.

**Comment 11:**

Line 222: 'these correlations are largely consistent within local regions', what do you mean here? It is meaningless to correlate production and harvested area, which is an obvious correlation.

**Response to Comment 11:**

We completely understand the your question about the expression in line 222, we will carefully revise this expression and its context, striving to accurately and concisely explain our research findings and their significance

1) As you pointed out, there is usually a significant linear correlation between production and harvested area, which has been widely recognized in the academic community. The expression in line 222 did not effectively convey our research basis and may cause confusion for readers. We apologize for this. In fact, what we want to emphasize is that, in addition to harvested area, production also has complex non-linear coupling relationships with multiple environmental factors, and these relationships exhibit significant regional differentiation characteristics in space. It is based on the recognition of this regional heterogeneity that this study adopts an agro-ecological zoning modeling strategy, striving to establish targeted production relationships in different regions.

2) Therefore, in order to more accurately and clearly convey the theoretical basis of this study, we will reorganize and restate line 222 and its context. The proposed revision is: **"Within each AEZ, crop production at the pixel level exhibits non-linear relationships with multiple environmental drivers. Moreover, these non-linear relationships vary significantly across different AEZs. This highlights the importance of developing zone-specific production allocation models to capture spatial heterogeneity."** In the revised manuscript, we will also supplement the theoretical explanation of the production composition mechanism in the literature review section, emphasizing the interactive influence

of natural conditions and artificial management; in the discussion section, we will further analyze the advantages and limitations of the agro-ecological zoning model and future improvement directions.

3) Thank you again for your valuable comments, which are of great benefit to improving the logical expression of our article.

**Comment 12:**

Line 279: you may not use comparative degree in the sentence when you accurately did not make comparison between two things. There are same grammar errors in the close following sentences.

**Response to Comment 12:**

1) We sincerely appreciate your careful review and pertinent criticism. We will more strictly adhere to English grammar rules in the revised manuscript. While correcting grammatical errors, we will also thoroughly check other language expression issues in the article, such as redundancy, repetition, and inappropriate wording, striving to make the writing more concise, fluent, and idiomatic.

2) To further improve the language quality of the article, we will invite native English-speaking peers to review and polish the revised manuscript.

**Comment 13:**

Line 284: 'regions --- have ---' is not good expression. I strongly suggest the authors rewrite the manuscript.

**Response to Comment 13:**

We sincerely appreciate your suggestion on the writing quality of the article. We fully accept your recommendation and will carefully rewrite and refine the entire manuscript. We will also invite native English-speaking peers to review and polish the revised manuscript.

**Comment 14:**

Line 290 and 295: it is very mess and redundant paragraph. Especially, at the result section, the authors talked a lot of potential implications which should be put into the discussion section, and by the way these implications also mentioned in the introduction and discussion sections too.

**Response to Comment 14:**

1) We sincerely appreciate your valuable comments on the article structure and content arrangement. We fully agree with your point of view that excessively discussing the research implications in the results section is indeed a serious problem that can disrupt the logical structure of the article and cause confusion for readers. We deeply apologize for any inconvenience caused by this. We will carefully revise lines 290, 295, and related paragraphs according to your suggestions.

2) At the same time, we will also check the entire manuscript to see if there are content repetition issues in sections such as the introduction, results, and discussion, especially regarding the description of research implications and significance. For unnecessary repetitive discussions, we will delete them; for necessary content, such as problem statements in the introduction and results interpretation in the discussion section, we will focus on how to express them in a more concise and accurate language, avoiding verbosity and redundancy.

**Comment 15:**

Line 305: this paragraph is to introduce the method, which should be put into the method section, and it is no need to introduce the common information as the reader easily know it.

**Response to Comment 15:**

1) We completely agree with the your point of view that placing method-related content in the results section is indeed not reasonable enough and will affect the logical structure and readability of the article. Thank you for your careful review and reminder. We will move line 305 and related paragraphs to the methods section in the revised manuscript to ensure the normative and completeness of the article structure. At the same time, we will also appropriately delete some of the content, removing some overly basic background knowledge introductions, and highlighting the key information descriptions of the methods used in this study.

2) Thank you again for your valuable suggestions. We will scrutinize the organization of the article with more rigorous standards, ensuring that the focus of each part is prominent, the details are appropriate, and the content is coherent, so that the revised manuscript better meets the publication requirements of the ESSD journal.

**Comment 16:**

Figures (fig. 2 and 3 at least) should show up after the main text that mentioned them.

**Response to Comment 16:**

We sincerely appreciate your careful examination of the correspondence between the figures and the main text in the article and for pointing out the issue of figures not being closely adjacent to the text. We will carefully check and adjust the positions of all figures in the revised manuscript to ensure that the figures are in the correct positions to help readers read and understand better.

**Comment 17:**

Line 309: I cannot read these numbers in fig. 3.

**Response to Comment 17:**

We sincerely appreciate your pointing out this issue. Regarding the figure and table analysis issues in line 309 and its context, we will systematically sort out the figure and table analysis paragraphs in the article, carefully check each numerical description, and ensure that they can all find direct basis in the corresponding figures and tables. We will also proofread the figure and table titles and legends to check the consistency of their content descriptions with the main text; verify that the symbols and abbreviations used in the figures and tables are all explained in the main text.

**Comment 18:**

Line 350-360: you may consider to improve the writing here.

**Response to Comment 18:**

We sincerely appreciate your valuable suggestions on the writing. Regarding the content of lines 350-360, i.e., the numerical analysis part of the "Evaluation of Model Performance in Different Regions" section, we will optimize the writing, sorting out the logical thread of data analysis, ensuring clear analysis ideas, enhancing the coherence of the context, and making the text expression more fluent and natural. At the same time, in order to further improve the overall language quality of the article, we will invite native English-speaking peer experts to polish the entire manuscript to meet the requirements of the ESSD journal.

**Comment 19:**

Section 3.3: this section is the best choice to examine the reasonability of the method. All four figures showed the most important role of location for simulating production, which is unreasonable. Why did the authors select the location as an input feature?

And why the location is important for simulating production. These results made me suspect this method. I really hope the authors be careful here, and the wrong method made the validation not too bad but will induce totally wrong regional or global distribution. Besides, the satellite-based features should play an important role, but they did not in this study. The explanation of this section does not make sense mostly. For example, at paragraph with line 380, and most of these explanations also are correct for other crops.

**Response to Comment 19:**

1)  We sincerely appreciate your valuable comments. After careful analysis, we realize that directly taking the average of the feature importance of each region on a global scale is indeed unreasonable. **This analysis approach may weaken the differences in crop planting conditions and model influencing factors between regions, and cannot objectively reflect the key features of different agro-ecological zones.** In fact, during the experiment, we also observed that for different regions and crop types, there are significant differences in the dominant factors affecting production estimation. Simply averaging the feature importance of all regions may obscure this heterogeneity, leading to the location feature being overly emphasized while the roles of other important features are underestimated.

2)  In response to the above problems, we will make major adjustments to the feature importance analysis method in the revised manuscript. Specifically, we will summarize the key features of the corresponding crop production estimation model for each region according to the agro-ecological zoning and crop type, and present them in the form of a heat map, as shown in **Figure 3**, **Figure 4**, **Figure 5** and **Figure 6**. By horizontally comparing the analysis results of different regions, we will more intuitively and accurately present the differences between regions, highlighting the specificity of different agro-ecological zones in the production composition mechanism. Due to the large number of indicators and regions, the readability of the figures below may be insufficient. We will optimize them in the revised manuscript to present them in a clearer way.

[Figure]

**Figure 3. Heatmap of feature importance for each modeling region: Maize.**

[Figure]

**Figure 4. Heatmap of feature importance for each modeling region: Wheat.**

[Figure]

**Figure 5. Heatmap of feature importance for each modeling region: Rice.**

[Figure]

**Figure 6. Heatmap of feature importance for each modeling region: Soybean.**

3) **Regarding the rationality of using location as a model input feature**, we will provide a more detailed explanation and discussion in the text. The choice to include location information in model training is mainly based on the following considerations: First, location features are not used to characterize the production differences between agro-ecological zones, but to represent the spatial association of different pixels within the agro-ecological zones. In the existing gridded

analysis framework, if location information is not introduced, the model will treat each pixel as an independent individual, ignoring the correlation between adjacent pixels in production composition. By introducing location features, the model can better learn and utilize the spatial autocorrelation between pixels, thereby improving the accuracy of production estimation. **Recent studies have also shown that tree-based ensemble learning algorithms (such as XGBoost) can effectively capture the spatial association information contained in location features** [1-3]. In the revised manuscript, we will appropriately increase the citation and discussion of these literatures to highlight the scientific and necessary nature of using location features. It should be noted that we are not trying to replace other environmental and management factors with location features, but rather use them as a supplement to improve the model's ability to characterize the spatial differentiation patterns of production from multiple dimensions. We believe that on the basis of comprehensive consideration of natural conditions and location associations, the role of satellite remote sensing and other factors will be more fully utilized.

[1] Li, Y., Zeng, H., Zhang, M., Wu, B., Zhao, Y., Yao, X., Cheng, T., Qin, X., and Wu, F.: A county-level soybean yield prediction framework coupled with XGBoost and multidimensional feature engineering, International Journal of Applied Earth Observation and Geoinformation, 118, 103269, https://doi.org/10.1016/j.jag.2023.103269, 2023.

[2] Mojaddadi, H., Pradhan, B., Nampak, H., Ahmad, N., and Ghazali, A. H. bin: Ensemble machine-learning-based geospatial approach for flood risk assessment using multi-sensor remote-sensing data and GIS, Geomatics, Natural Hazards and Risk, 8, 1080–1102, https://doi.org/10.1080/19475705.2017.1294113, 2017.

[3] Papacharalampous, G., Tyralis, H., Doulamis, A., and Doulamis, N.: Comparison of Tree-Based Ensemble Algorithms for Merging Satellite and Earth-Observed Precipitation Data at the Daily Time Scale, Hydrology, 10, 50, https://doi.org/10.3390/hydrology10020050, 2023.

4) **Regarding the problems in the explanation part of Section 3.3,** we will carefully sort out the existing expressions, find and correct the unreasonable or logically confusing arguments. At the same time, we will also pay more attention to the interpretation of the results of different crops, deeply analyzing the reasons for the model's performance differences in different crops, and improving the pertinence and effectiveness of the explanations. We strive to accurately convey the applicable conditions and limitations of the model through the optimization and improvement of the explanation content, providing necessary references for readers to comprehensively and objectively understand the research results.

**Comment 20:**

3.4 Comparing with Existing Datasets: I don't think this dataset is consistent with other existing datasets. Like fig. 9 showed the large difference compared to SPAM2010 for all four crops. It can also be found the systematic differences from fig. 10.

**Response to Comment 20:**

We sincerely appreciate your questioning on the data comparison results. We will focus on improving the data comparison section in the revised manuscript, striving to present a more comprehensive and clear data quality assessment result to the readers.

■ **Regarding the comparison results with SPAM 2010:**

1) We indeed did not effectively show the consistency level between the two datasets in Figure 9 of the preprint. Limited by the form of the scatter plot, it is difficult for readers to fully understand the distribution density of the two datasets in different value ranges, which may overestimate the degree of difference between them. In order to more intuitively and accurately present the actual differences between GGCP10 and SPAM 2010, we performed pixel-by-pixel difference calculations on the two datasets and plotted a frequency distribution histogram of the differences (as shown in **Figure 7**).

[Figure]

**Figure 7. Frequency distribution histogram of pixel differences between GGCP10 and SPAM 2010: (a) Maize; (b) Wheat; (c) Rice; (d) Soybean.**

2) From the histogram, it can be seen that the pixel differences between the two datasets are mostly concentrated in the range of -0.89 kilotons to 0.60 kilotons. For maize, wheat, rice, and soybean, the percentage of pixels falling within this range are 60.7%, 57.1%, 55.6%, and 74.0%, respectively. Further statistics show that the percentage of pixels with differences between -1 and 1 reaches 66.4%, 64.5%, 60.3%, and 78.2%, respectively. Although there are still a small number of pixels with absolute difference values greater than 5.00 kilotons, their proportions in the four crops are relatively small, at only 8.3%, 5.3%, 13.2%, and 2.5%, respectively. Combining the above results, it can be seen that despite non-negligible local differences, GGCP10 still has a high degree of consistency with SPAM 2010 globally and in major regions.

3) **Moreover, we will provide more details for evaluating the consistency and difference between GGCP10 and SPAM 2010, as shown in Figure 8.** We selected key regions such as Africa (maize), Western Europe (wheat), Southeast Asia (rice), and Brazil and Argentina in South America (soybean). Due to the

differences in the definition of arable land between the two datasets, inconsistencies in the covered pixels are inevitable. From a detailed perspective, the two datasets have very high consistency in high-production regions, while inconsistent regions are mainly located in low-value areas. Additionally, compared to SPAM2010, GGCP10 exhibits smoother spatial transitions.

[Figure]

**Figure 8. Spatial comparison of crop production between SPAM 2010 and GGCP10 datasets for selected regions: (a) Maize production in Africa; (b) Wheat production in Western Europe; (c) Rice production in Southeast Asia; (d) Soybean production in Brazil and Argentina, South America.**

4) In the revised manuscript, we will use more appropriate charts to provide readers

with a more accurate and detailed perspective for difference assessment. At the same time, we will also appropriately increase the quantitative description and discussion of consistency levels in the main text to further highlight the overall reliability of GGCP10.

■ **Regarding the comparison with AsiaRiceYield4km**

1) We believe your comments are very pertinent. GGCP10 does show a high degree of consistency with AsiaRiceYield4km in overall trends and patterns of change, which can be seen from the correlation coefficients (0.91-0.93) and coefficients of determination (0.83-0.86) between the two datasets. However, as revealed in Figure 10 in the preprint, GGCP10 does exhibit a certain degree of systematic overestimation or underestimation in some high-value regions relative to AsiaRiceYield4km.

2) There may be multiple reasons for this local systematic bias. Among them, the difference in spatial resolution between the two datasets cannot be ignored. The original resolution of AsiaRiceYield4km is 4km, while GGCP10 is 10km. For comparison purposes, we resampled the former to a 10km resolution. During the downscaling process, local high and low value differences may be smoothed to a certain extent, resulting in a weakened gradient of change in some regions of the resampled AsiaRiceYield4km. On the other hand, there are also differences in the algorithms and models used by the two datasets in their ability to characterize local production heterogeneity.

3) In addition, differences between the two datasets in terms of training sample representativeness, data source quality, mixed pixel processing, etc., may also introduce systematic biases in local regions. In the revised manuscript, we will use spatial analysis methods to meticulously characterize the local difference patterns between GGCP10 and AsiaRiceYield4km, focusing on regions where the two datasets show significant deviations in high-value areas. We will visually present the spatial distribution of these regions through difference maps, scatter density plots, and other means, and discuss the possible causes of these systematic biases.

■ **For the comparison with India DES and USDA data**

We will also refine and improve the comparison with India DES and USDA data in the revised manuscript. In addition to showing global consistency indicators, we will also select typical regions to focus on analyzing the degree of agreement between GGCP10 and validation data at the local scale.

1) For example, in Figures 9, 10, 11 and 12, we showcase the spatial distribution of four crops at the county scale in 2010, 2015, and 2020 for both USDA survey data and GGCP10. The results demonstrate that the two datasets exhibit a high level of overall consistency. However, in some regions with lower production, the consistency is relatively poor. One possible reason for this discrepancy is that our data undergoes a consistency processing based on statistical data, while the

USDA data is derived from surveys. The USDA survey data may not fully align with the official statistical data due to factors such as sampling errors, differences in statistical calibers, and data collection methods. This can lead to inconsistencies between the two datasets, particularly in regions with lower production where the impact of these factors may be more pronounced. We'll discuss more details in a revised manuscript.

2) By providing a more in-depth analysis of the agreement between GGCP10 and USDA data at the local scale, we aim to offer readers a clearer understanding of the strengths and limitations of our dataset. This information will be valuable for users when applying GGCP10 data in different regions and production scenarios. Furthermore, by identifying areas where consistency could be improved, we can guide future efforts to refine our data processing methods and enhance the overall reliability of GGCP10.

[Figure]

**Figure 9. Spatial comparison of Maize production between USDA survey data and GGCP10 at the county level for the years 2010, 2015, and 2020.**

[Figure]

**Figure 10. Spatial comparison of Wheat production between USDA survey data and GGCP10 at the county level for the years 2010, 2015, and 2020.**

[Figure]

**Figure 11. Spatial comparison of Rice production between USDA survey data and GGCP10 at the county level for the years 2010, 2015, and 2020.**

[Figure]

**Figure 12. Spatial comparison of Soybean production between USDA survey data and GGCP10 at the county level for the years 2010, 2015, and 2020.**

Thank you again for your valuable comments. In the revised manuscript, we will follow your suggestions to provide more detailed quantitative indicators and graphical explanations in the dataset comparison, and conduct a more comprehensive discussion on data limitation issues. We sincerely hope that these revisions can more comprehensively and forcefully demonstrate the advantages and shortcomings of GGCP10, providing an objective and transparent reference for data users and research peers. We also sincerely invite you to review the revised manuscript once we submit it and kindly request your continued valuable comments and suggestions.

---

## Author Comment (AC5)

We sincerely appreciate your affirmation of this research and your valuable suggestions. As you pointed out, continuous and accurate mapping of crop production at different spatial and temporal scales is of great significance for agricultural production monitoring and food security early warning. The release of GGCP10 is precisely intended to provide crop production data with high spatiotemporal resolution for scientists and practitioners engaged in related research.

Regarding the issue of model training sample representativeness, we fully agree with the point you raised. Due to significant differences in topography and climatic conditions between countries, even within a single country, crop production levels may vary considerably between different regions. FAO's national-scale statistical data can hardly reflect this spatial heterogeneity at the regional scale. However, it should be noted that in this study, we did not directly use national average production data to train the model. Instead, we constructed the association between pixel-level production and multi-source data at the agro-ecological zone scale based on the gridded production data of the reference year, and predicted the gridded production of the target year based on this. In the revised manuscript, we will clarify our modeling approach and provide a more thorough discussion of the data representativeness issue mentioned above.

In fact, we compared the dataset with the SPAM 2010 dataset, which is a globally-covered dataset. In the revised manuscript, we will introduce in more detail the consistency of our dataset with the SPAM 2010 dataset in different regions, as shown in **Figure 1**. In the future, we will also actively integrate crop production data products from other sources and further improve the regional applicability of GGCP10 through cross-validation of multi-source data. At the same time, we also welcome more researchers to validate GGCP10 using the ground observation data they possess, promoting the continuous improvement of data quality through extensive academic exchanges.

[Figure]

**Figure 1. Spatial comparison of crop production between SPAM 2010 and GGCP10 datasets for selected regions: (a) Maize production in Africa; (b) Wheat production in Western Europe; (c) Rice production in Southeast Asia; (d) Soybean production in Brazil and Argentina, South America.**

Finally, thank you very much for your suggestion on incorporating factors such as agricultural inputs. In the current version of GGCP10, due to the lack of unified global agricultural input data, we find it difficult to directly use agronomic management measures for modeling. This is indeed a limitation of the current research. It should be pointed out that although there is a lack of direct agronomic management data, some of the factors included in the model can, to a certain extent, reflect the spatial differences in agronomic management levels. For example, irrigation data reflects differences in irrigation management inputs, which greatly

influence crop growth and production. Crop planting area data partially reflects farmers' planting preferences and resource allocation decisions for different crops, demonstrating farmers' responses to market conditions and policies.

Some of the remote sensing-derived indicators we included can also reflect the impact of agronomic management to a certain degree. For instance, the Vegetation Condition Index (VCIx) describes the historical relative level of vegetation conditions during the study period. A higher VCIx indicates relatively better crop growth during that period, which to some extent benefits from farmers' good field management. Indicators such as Net Primary Productivity (NPP) and Leaf Area Index (LAI) reflect crop biomass accumulation and photosynthetic intensity. Higher NPP and LAI are often the result of good management.

We will provide further explanation and analysis of this issue in the discussion section of the revised manuscript.

Thank you again for your insightful comments.